# Discovery of Compounds That Selectively Repress the Amyloidogenic Processing of the Amyloid Precursor Protein: Design, Synthesis and Pharmacological Evaluation of Diphenylpyrazoles

**DOI:** 10.3390/ijms232113111

**Published:** 2022-10-28

**Authors:** Christophe Mesangeau, Pascal Carato, Nicolas Renault, Mathilde Coevoet, Paul-Emmanuel Larchanché, Amélie Barczyk, Luc Buée, Nicolas Sergeant, Patricia Melnyk

**Affiliations:** 1U1172—LilNCog—Lille Neurosciences & Cognition, Univ. Lille, Inserm, CHU Lille, F-59000 Lille, France; 2U1286—INFINITE—Institute for Translational Research in Inflammation, Univ. Lille, Inserm, CHU Lille, F-59000 Lille, France

**Keywords:** Alzheimer’s disease, amyloid protein precursor, lysosome, autophagy, β-secretase, pyrazole

## Abstract

The rationale to define the biological and molecular parameters derived from structure–activity relationships (SAR) is mandatory for the lead selection of small drug compounds. Several series of small molecules have been synthesized based on a computer-assisted pharmacophore design derived from two series of compounds whose scaffold originates from chloroquine or amodiaquine. All compounds share similar biological activities. In vivo, Alzheimer’s disease-related pathological lesions are reduced, consisting of amyloid deposition and neurofibrillary degeneration, which restore and reduce cognitive-associated impairments and neuroinflammation, respectively. Screening election was performed using a cell-based assay to measure the repression of Aβ_1–x_ peptide production, the increased stability of APP metabolites, and modulation of the ratio of autophagy markers. These screening parameters enabled us to select compounds as potent non-competitive β-secretase modulators, associated with various levels of lysosomotropic or autophagy modulatory activities. Structure–activity relationship analyses enabled us to define that (1) selectively reducing the production of Aβ_1–x,_ and (2) little Aβ_x–40/42_ modification together with (3) a decreased ratio of p62/(LC3-I/LC3-II) enabled the selection of non-competitive β-secretase modulators. Increased stability of CTFα and AICD precluded the selection of compounds with lysosomotropic activity whereas cell toxicity was associated with the sole p62 enhanced expression shown to be driven by the loss of nitrogen moieties. These SAR parameters are herein proposed with thresholds that enable the selection of potent anti-Alzheimer drugs for which further investigation is necessary to determine the basic mechanism underlying their mode of action.

## 1. Introduction

Alzheimer’s disease (AD), the most frequent form of dementia worldwide, is a brain disease associated with two neuropathological processes, leading to a slow decline in cognitive and behavioral disabilities. Neuropathological lesions include neurofibrillary tangles (NFTs) and amyloid deposits, together with astrogliosis, neuroinflammation, and neuronal death [1]. NFTs are made up of the intraneuronal accumulation and aggregation of abnormally modified isoforms of the microtubule-associated Tau. Parenchymal amyloid deposits are composed of amyloid-β (Aβ) peptides originating from complex and sequential proteolytic cleavages of a precursor protein, namely, the amyloid precursor protein (APP) (for a comprehensive review see [2]). The non-amyloidogenic pathway, initiated by the α-secretase cleavage of APP, is opposed to the amyloidogenic pathway that is initiated by a primary β-secretase cleavage at the first amino acid of the Aβ peptide sequence (Figure 1) [3,4]. Both secretase-mediated steps shed soluble ectodomains of APP (sAPPα and sAPPβ) and membrane-bound carboxyl-terminal fragments (APP-CTFs), referred to as αCTF and βCTF. These αCTFs or βCTFs are further cleaved by the γ-secretase to give rise to p3 and Aβ peptides, respectively, along with the APP intracellular domain (AICD). Along the cell-secretory pathway, APP can also be cleaved by β-secretase at the β′-site at position 11 of the Aβ peptide sequence, which is suggested to be protective [5,6,7,8]. The definite diagnosis relies on the presence of both NFTs and amyloid deposits, which are therefore considered to stem pathophysiological processes leading to AD, and which posit that therapeutic development should therefore target both of these processes.

AD treatment remains symptomatic and currently, disease-modifying treatment remains ill-defined except for aducanumab immunotherapy, which reduces the amyloid load and Tau PET imaging in clinical trials [9]. The definite diagnosis relies on these pathological processes, which are therefore all related to AD pathophysiology. Therefore, alternative disease-modifying therapeutic options are urgently needed, and over the past few years, much effort has been dedicated to the development of such disease-modifying drugs [10]. One such strategy would be the development of drugs that efficiently modify both amyloid and Tau pathological processes.

Redirection of APP processing can be achieved either by blocking the amyloidogenic pathway or by promoting the non-amyloidogenic pathway. The results of these activities would be a decrease in Aβ secretion and an increase in αCTF. We previously showed that CQ inhibits Aβ production, whereas levels of other APP metabolites such as APP-CTFs and AICD are maintained and even increased, whereas the γ-secretase cleavage of Notch remains unmodified, precluding the contribution of the γ-secretase in this sequential proteolytic process of APP [11,12].

We have recently developed two families of compounds derived from either CQ (family A) or amodiaquine (AQ, family B) (Figure 2) that demonstrated a strong inhibitory effect on both Aβ_1–40_ and Aβ_1–42_ secretions [13,14,15,16,17,18,19]. These effects were associated with a strong increase in αCTFs and AICD levels. These families of molecules interact with the autophagic/endolysosomal systems, some of which have been shown to be effective against both amyloid and Tau pathologies in vivo [14,17]. The lead compound of family A is currently in a phase II clinical trial for the treatment of tauopathy progressive supranuclear palsy [20]. To try to identify new and more efficient compounds in the absence of a defined molecular target of the previously developed compounds, a ligand-based computer-assisted pharmacophore modeling approach [21], coupled with a de novo design, was implemented. In the first paper, we described a new family of compounds based on a biaryl scaffold decorated with amino side chains (family C, Figure 2) [22]. Among these compounds, several were shown to inhibit Aβ_1–x_ peptide secretion and to promote αCTFs and AICD stabilities as the parent compounds (families A and B). In this previous structure–activity relationship based on the analysis of APP metabolism and markers of autophagy, non-competitive β-secretase inhibitors with or without a lysosomotropic activity were identified [22]. Interestingly, compound PEL24–199 (compound **31** in [22]) with *n* = 3, *m* = 2, X = CH, a phenyl ring, Ar, and a dimethylamino group, NR_1_R_2_, provided the most interesting pharmacological properties. In vivo, this compound, with reduced lysosomotropic activity, reduced neurofibrillary degeneration, astrogliosis, neuroinflammation, and short-term spatial memory impairments in the Thy-Tau22 transgenic mouse model of hippocampal tauopathy [23]. Our results suggest that the non-competitive β-secretase inhibition is necessary for the pharmacological effect in vivo whereas the lysosomotropic activity should be reduced and likely dispensable for drug efficacy.

In this paper, we describe a new family of compounds derived from a diphenylpyrazole scaffold decorated with amino side chains (family D, Figure 2). In the screening process, the assessment of APP metabolites including soluble APP fragments arising from the β- or α-secretase endoprotease activity, the so-called sAPPβ and sAPPα, together with the carboxy-terminal APP fragments (APP-CTFs) and Aβ_1–X_ peptides, provide outcomes on non-competitive β-secretase inhibition [22]. The lysosomotropic activity of our compounds was evaluated by a lower degradation rate of CTFα, AICD, and LC3-I/LC3-II, and p62/SQSTM1(Sequestresome protein 1) markers of autophagy whose expression and ratio increase as the result of autophagy to lysosome flux inhibition.

## 2. Results

### 2.1. Ligand-Based Pharmacophore Modeling

We previously described a computer-assisted ligand-based approach and applied this method to determine a common pharmacophoric model between the two families of compounds A and B considering the sole inhibitory effect on Aβ production (Figure 1) [22]. In brief, a set of 51 compounds of family A [14] and 35 of family B [18,19] were chosen regarding the ability of the compounds to inhibit Aβ_1–x_ secretion. Compounds **A1** and **B1** are representative compounds of families A and B, respectively. The best 3D pharmacophoric model, describing four spatial points Positive/Positive/Positive/Aromatic (PPPA), was used to design new APP modulators. Several structures were then conceptualized to spatially organize pharmacophoric elements in agreement with this model. We then chose chemical structures based on a scaffold enabling the orientation of amino side chains in three different orientations. Two different scaffolds were first obtained: (1) a biaryl scaffold (family C) with amino side chains previously described [22], and (2) a diphenylpyrazole scaffold (family D) decorated with amino side chains.

This family of compounds was designed around a pyrazole scaffold substituted at the 1- and the 3-positions by phenyl rings, each of which being optionally further substituted by R_1_ and R_2_ groups, respectively. Two series of compounds were designed with different substituents at the 4-position of the pyrazole moiety, one with a methyl substituent and the other one with an (*N*,*N*-dimethylamino)alkyl group substituent wherein the alkyl is methyl or propyl. In both series, the optional R_1_ and R_2_ substituents of the phenyl rings were independently chosen from (dialkylamino)alkyl and (heterocyclylamino)alkyl. The nature of the amino side chains on the pyrazole scaffold was chosen based on previous biological readouts. Different modulations then allowed us to select and validate the importance of adding two or three different amino side chains: monoamine, linear, or cyclic diamine.

The compounds were then tested to evaluate the impact of the structural modifications on cytotoxicity and metabolism of APP (APP-CTF quantification, secretion of Aβ_1–x_ peptides, ...) using SY5Y-APP^wt^ cells, a well-established model mastered in our laboratory and used to evaluate and compare compounds of all family of drugs synthesized thus far. sAPPα and sAPPβ were also quantified. In addition to Aβ_1–40_ and Aβ_1–42_, which arise from the canonical β-secretase and γ-secretase, Aβ_x–38/40/42_ peptides were also measured since truncated Aβ species are also produced such as Aβ_11–x_, which are generated by the β-secretase along the secretory pathway (Figure 1) [5].

### 2.2. Synthetic Chemistry

For the synthesis of the first series with (*N*,*N*-dimethylamino)alkyl groups in the 4-position of the pyrazole scaffold of compounds **24**–**35**, we first describe the preparation of intermediates **10**–**11**, **14**–**15** (Figure 1).

Arylhydrazines **3** and **4** were prepared from anilines **1** and **2** by diazotization in hydrochloric acid (37%), followed by the reduction of the diazonium salts using tin(II) chloride according to the reaction protocols known in the literature [24,25,26]. Compound **5** was prepared to start from 1-[4-(3-bromopropyl)phenyl]ethanone and dimethylamine (2 M in methanol). Compound **5** was then condensed with **3**–**4** to provide hydrazone derivatives **6**–**7**. By the double addition of Vilsmeier–Haack [27,28], the desired cyclized aldehydes **8**–**9** were obtained. Starting from derivatives **8**–**9**, two routes were used to afford the final compounds **10**–**11**, **14**–**15** with [(*N*,*N*-dimethyl)amino]methyl or [(*N*,*N*-dimethyl)amino]propyl substituents at the 4-position of the pyrazole scaffold. The first route was a reductive amination of ***8***–**9** with sodium triacetoxyborohydride, dimethylamine, and acetic acid in DCE to give compounds **10**–**11**. The second route, in two steps, was a Horner–Wadsworth–Emmons olefination of **8**–**9** with sodium hydride, diethylcyanomethylphosphonate in THF, followed by catalytic hydrogenation of the intermediates in the presence of Raney nickel and palladium on carbon in methanol saturated with ammonia under a hydrogen atmosphere, to afford compounds **12**–**13**. The last step was the reductive methylation of derivatives in methanol with formaldehyde, and sodium triacetoxyborohydride to afford the corresponding compounds **14**–**15**.

Starting from compounds **10**–**11** and **14**–**15**, final compounds **24**–**35** were prepared in three steps (Figure 2).

Reduction of the ester group of compounds **10**–**11**, **14**–**15** with lithium aluminum hydride in THF gave derivatives **16**–**19**, which were then oxidized with manganese oxide in chloroform to the corresponding compounds **20**–**23** (Figure 2). Subsequently, target compounds **24**–**35** were prepared by reductive amination of **20**–**23** with the appropriate amine in the presence of sodium triacetoxyborohydride and acetic acid in DCE.

For the synthesis of the second series with a methyl group in the 4-position of the pyrazole scaffold (compounds **61**–**72**), we first describe the preparation of intermediates **44**–**45** (Figure 3).

Compound **4** and 4-or 3-cyanoacetophenone (**36**–**37**) were condensed in methanol to provide hydrazone derivatives **38**–**39**, which were then treated with Vilsmeier–Haack reagent with POCl_3_ in DMF to afford pyrazoles **40**–**41**. Selective reduction of the aldehyde functions (**40**–**41**) with triethylsilane and trifluoroacetic acid gave nitrile derivatives **42**–**43**, which were reduced with the borane–THF complex in THF to afford compounds **44**–**45**.

Starting from intermediates **44**–**45**, compounds **61**–**72** were prepared according to two routes (Figure 4).

Compounds **50**–**51**, with a [(*N*,*N*-dimethyl)amino]methyl group, R_2_, were synthesized in three steps starting from derivatives **44**–**45**, which were engaged in the reductive methylation with formaldehyde, sodium triacetoxyborohydride in methanol to furnish compounds **46**–**47**. Using the previously described method in Figure 2, the reduction of the ester group of compounds **46**–**47** with lithium aluminum hydride in THF gave derivatives **48**–**49**, which were then oxidized with manganese oxide in chloroform to afford aldehydes **50**–**51**.

Compounds **59**–**60**, with a [3-(dimethylamino)propyl-methyl-amino]methyl group, R_2_, were synthesized in four steps starting from derivatives **44**–**45**, which were coupled in DCM with EDCI, HOBt, and 3-(dimethylamino)propionic acid **52**, previously prepared to start from beta-alanine, formic acid, and formaldehyde (37%) in water. The obtained compounds **53**–**54** were then reduced with lithium aluminum hydride and aluminum chloride to afford derivatives **55**–**56**, followed by methylation of the secondary amine with formaldehyde, and sodium triacetoxyborohydride in methanol under reductive alkylation conditions to give compounds **57**–**58**. The last step was an oxidation reaction with manganese oxide in chloroform and aldehydes **59**–**60** were obtained.

Finally, target compounds **61**–**72** were prepared by reductive amination of **50**–**51**, **59**–**60** with the appropriate amine in the presence of sodium triacetoxyborohydride.

### 2.3. Effect of Compounds on APP Metabolism

The final and selected intermediate compounds were evaluated for their ability to modulate APP processing in the SH-SY5Y human neuroblastoma cell line stably expressing the neuronal isoform of human wild-type APP^695^ (SY5Y-APP^WT^), a well-established model for the study of APP metabolism. APP carboxy-terminal fragments (αCTFs and AICD) levels were assessed by Western blotting (WB). The effect of the reference and test compounds on αCTFs and AICD is expressed as the intensity of corresponding WB bands [14]. The concentration of compounds able to increase 20-times αCTFs and AICD (C_20_) when compared to the control condition was calculated. Selected compounds were also evaluated for their ability to modulate the secretion of soluble APP fragments (sAPPα and sAPPβ). Aβ_1–x_ levels (Aβ_1–40_ and Aβ_1–42_) after treatment with the reference and tested compounds were measured in the cell media by ELISA (Table 1). The results are expressed as IC_50_ values, which correspond to the concentration of a given compound that inhibits Aβ_1–x_ concentration by 50% (either Aβ_1–40_ or Aβ_1–42_) in comparison to the concentration of Aβ_1–x_ in non-treated SY5Y-APP^WT^ cells. Selected hit compounds were then further evaluated for their ability to decrease the level of N-truncated Aβ peptides (Aβ_x–38_, Aβ_x–40_, and Aβ_x–42_).

For the first series of compounds **16**–**35**, different modulations were applied with the R_1_ group, in the meta- or para-position of the phenyl ring, as hydroxymethyl, [(*N*,*N*-dimethyl)amino]methyl or [(4-methyl)piperazin1-yl]methyl or [3-(dimethylamino)propyl-methyl-amino]methyl. Intermediates **16** and **17**, with a hydroxyl substituent as the R_1_ group, were tested to evaluate the influence of the presence of an additional positive charge or not. Substitution with an R_1_ hydroxyl group led to an increase in cell toxicity (Table 1), whereas substitution with a [(*N*,*N*-dimethyl)amino]methyl or [(4-methyl)piperazin1-yl]methyl or [3-(dimethylamino)propyl-methyl-amino]methyl group reduced the cell toxicity to concentrations above 100 µM. First, when considering the effect on the Aβ_1–X_ levels, the variation in the position of the R_1_ group gave a micromolar IC_50_ close to reference **A1** and lower than reference **B1** and **CQ** for compounds **24**–**31** (Table 1). The compounds with [(*N*,*N*-dimethyl)amino]methyl or [(4-methyl)piperazin1-yl]methyl, as the R_1_ group, showed the best efficacy toward Aβ_1–40_ and Aβ_1–42_ (Aβ_1–40_: IC_50_ = 1.5–4.0 µM and Aβ_1–42_: IC_50_ = 2.4–5.8 µM) compared to derivatives with [3-(dimethylamino)propyl-methyl-amino]methyl substitution **32**–**35** and benzylic alcohols **16** and **17** (Aβ_1–40_: IC_50_ = 4.2–12.4 µM and Aβ_1–42_: IC_50_ = 5.4–13.2 µM). (Table 1). Second, αCTFs and AICD C_20_ were evaluated and shown to be lower for compounds **24**–**32** than for **33**–**35** and **16**–**17**, showing on average a C_20_ also lower than the reference **A1**, **B1**, and **CQ** (Table 1), together suggesting that the increased stability of CTFα and AICD is indicative of a gain in lysosomotropic activity, similar to CQ (C_20_ for CTFα and AICD over 10 µM). In general, the para position yields a better reduction of Aβ_1–x_ concentrations in contrast to the little impact of CTFα and AICD C_20_. Considering the pyrazole ring, a shorter chain (*n* = 1) yielded lower IC_50_ and C_20_ than a longer chain (*n* = 3), especially when R_1_ is the [3-(dimethylamino)propyl-methyl-amino]methyl group. Compounds **24**–**35** present very low cytotoxicity (CC_50_ > 100 µM). In this first series of compounds, the replacement of the R_1_ amino chain with an uncharged CH_2_OH group yielded a significant increase in cytotoxicity and a low increase in IC_50_ for Aβ secretion, while αCTF and AICD C_20_ remained in the same range. This result underlines the importance of the presence of a positive charge in this *N*1 phenyl ring.

For the second series of compounds **61**–**72**, different modulations were performed within the R_1_ group at the meta-position of the *N*1 phenyl ring as well as the variation in the position (meta and para) of the R_2_ group as [(*N*,*N*-dimethyl)amino]methyl or [3-(dimethylamino)propyl-methyl-amino]methyl, and the introduction of a methyl group in the 4-position of the pyrazole scaffold, leading to very different results in terms of the cytotoxicity (CC_50_ between 2 and >100 µM). In general, compounds of this second series are more cytotoxic than those of the first, except for compounds **64**–**65** and **57**, which had a cytotoxicity CC_50_ above 100 µM, which is less toxic than the CQ CC_50_ in our conditions (CQ CC_50_ = 30 µM). On the other hand, the IC_50_ of Aβ secretion was homogeneous regardless of whether it was R_1_ or R_2_ and its position (Aβ_1–40_: IC_50_ = 1.2–3.9 µM and Aβ_1–42_: IC_50_ = 1.6–4.1 µM). C_20_ values for αCTF and AICD were homogeneous or even slightly lower than the first series of compounds, with compounds **69**, **71,** and **72** showing the best C_20_ values between 600 nM and 1 µM. In this series, the nature of the R_1_ group did not lead to any significant difference in APP metabolism. In contrast, the replacement of the R_1_ alkylamino chain with an uncharged CH_2_OH group yielded a significant decrease in activity (IC_50_ for Aβ secretion higher than 10 µM for **48**, αCTF higher than 10 µM for **57**). The presence of a diamino group such as R_2_ led to slightly improved values even if insignificantly different.

A selection of compounds with several selected activities was performed including sAPPα and sAPPβ, which were further explored. We selected compounds **67**, **69**, and **71** as the most efficient compounds decreasing Aβ production (IC_50_ around 1 µM) and increasing CTFα and AICD (C_20_ around 1 µM) but with various cytotoxic CC_50_. As a comparison, we also selected compounds **33** and **34** from the first series with lower efficacy in repressing Aβ_1–x_ production (IC_50_ comprised between 6 to 9 µM) and various effects on the CTFα and AICD expression (C_20_ around between 1 and 6 µM). The decrease in Aβ could result from a decrease in either β- or γ-secretase cleavage or an increase in the α-secretase cleavage of APP. Thus, medium concentrations of sAPPβ and sAPPα, which are the β- and α-secretase extramembrane shed APP fragments, were determined after cell drug treatments. All compounds were shown to decrease sAPPβ efficiently (Table 2). Interestingly, the greater the effect of the compounds on the repression of Aβ_1–x_, the lower the media concentrations of sAPPβ. The IC_50_ values of sAPPβ were thus greater for compounds **33** and **34** than for **67**, **69**, and **71** (Table 2). Conversely, compounds **67** and **71** were able to significantly increase the sAPPα concentrations (C_1.5_ around 1 µM), whereas the three other compounds did not show an increase in APPsα up to 10 µM (5 µM for compound **69**). These results strongly suggest that these compounds, in particular compounds **67** and **71**, reduce Aβ_1–x_ peptide secretion by repressing the β-secretase cleavage of APP while increasing the secretion of sAPPα, as suggested for other compounds from a previous family [22].

### 2.4. Effect of Compounds on Autophagic Flux and Lysosomal Degradation Pathways

Previous work dedicated to compounds of family C [22] has underlined the effect of our compounds on autophagic flux and the lysosomal degradation pathways. More precisely, the role of the amino side chains and their number was highlighted. Therefore, we assessed whether the activity of compounds involved the modulation of lysosomal activities toward the accumulation of APP-CTFs and AICD. Thus, we studied the effect of a selection of compounds on the autophagic flux, more particularly, we evaluated the effect of compounds **24**, **33**, **34**, **57**, **65**, **67**, **70**, and **71** on two markers associated with autophagy: p62 and LC3-I lipidation into LC3-II. To the previous selection of efficient compounds decreasing Aβ_1–x_ production and increasing CTFα and AICD, we added compound **65** with no cytotoxicity at 100 µM. As a comparison, we also selected compound **57** with a hydroxymethyl substituent as a control. In addition to CQ, bafilomycin A, a well-known inhibitor of autophagic flux, was used as a control. Treatment with compounds **67** and **71** induced an increase in p62 expression by 2.5 and 4-fold, respectively, compared to the control condition (Figure 3). LC3-I lipidation into LC3-II was also increased by 5.2 and 6.5-fold, respectively. An increase in the two markers was also observed with compound **57**, but to a smaller extent (1.7-fold for p62 and 2-fold for LC3-I/LC3-II). In contrast, compound **34** showed no significant effect on these two autophagy markers.

Pyrazole compounds are derived from a pharmacophore of structural superposition of families A and B of molecules themselves derived from CQ and AQ. Both CQ and AQ share alkalizing and lysosomotropic activities related to their accumulation in cell vesicular acidic compartments and polynitrogen protonability, making them weak bases. This latter chemical property could contribute to the understanding of structure–activity differences between compounds **34**, **57**, **67**, and **71**, associated with APP metabolism and autophagy flux modulatory function. We addressed this question by calculating the pKa of each nitrogen of compounds **34**, **57**, **67,** and **71** [29]. Compound **71** has four protonable nitrogens, although the second nitrogen of the R_1_ piperazine moiety has a low pKa. Three protonable nitrogens were found in compounds **34** and **67**, whereas compound **57** only had two. No relationship was observed between the pKa of compounds with either APP metabolism, the autophagy flux, or cell cytotoxicity.

We next assessed the potential relationship between the autophagy flux and Aβ_1–x_ production or the compound’s cytotoxicity.

While considering the compounds’ cytotoxicity, p62 expression was best correlated with this cytotoxicity while LC3 or the p62/LC3 ratio was not. Therefore, for the selection of lead compounds, these parameters of p62/LC3 ratio and p62 expression should be considered with regard to the Aβ_1–x_ production inhibitory activity and cell cytotoxicity. Accordingly, compound **24** from the first series and compounds **65** or **70** from the second appeared as having the selected pharmacological properties. Noticeably, these selected molecules had similar CC_20_ (µM) activities toward APP metabolism and αCTF and AICD expression.

Compounds from both series were selected to first determine the potential relationship between the repression of Aβ_1–x_ production and the increased expression of LC3 and p62. Noticeably, LC3 and p62 were not modulated within similar amplitudes (Figure 3). For instance, p62 was increased by 1.5-fold while LC3 remained unchanged for compound **33**, whereas LC3 was increased by 6-fold and p62 by 3-fold by compound **70**. We then also considered the ratio between the p62 and LC3 expression levels. While considering LC3 or p62 separately, the repression of Aβ_1–x_ was neither associated with LC3 nor the sole expression of p62. In sharp contrast, when considering the p62/LC3 ratio, the lower the ratio, the higher the repression of Aβ_1–x_. Both compounds **34** and **35**, with a ratio close to 1, had the lowest inhibitory activity (Table 3). Compounds **70** and **71**, having a p62/LC3 ratio of 0.30 and 0.34 had the strongest Aβ_1–x_ production inhibition (Table 3). These ratios are comparable with those of chloroquine and bafilomycin A1, both of which are well-known to repress Aβ production. This effect appeared to be independent of the p62 or LC3 expression levels, for instance, compound **24** had a p62/LC3 ratio of 0.49 whereas the LC3 and p62 expression levels were both lower when compared to chloroquine, bafilomycin A1, or compounds **70** and **71**.

## 3. Discussion

Previous studies in our laboratory underlined the effect of compounds derived from chloroquine CQ and amodiaquine AQ, two antimalarial compounds with lysosomotropic activities that are able to modulate the amyloid and Tau pathologies, both of which are the two major pathophysiological processes of AD. Two families of compounds (A and B, Figure 2) have been developed and used for a ligand-based approach. In a first study dedicated to the development of biaryl compounds (family C, Figure 2), we identified PEL24–199 (compound **31** in [22]), a compound able to inhibit Aβ_1–x_ peptide production without modifying Aβ_x–40/42_ and with little modulatory activity of the expression of αCTF and AICD when compared to the effect of chloroquine. This selective modulatory effect toward Aβ_1–x_ reduced the production, and maintenance of the global Aβ_x–40/42_ levels has been reported consequently to BACE1 overexpression or rare inherited mutations of APP, suggested by a modification of the enzyme/substrate recognition [30,31]. In vivo, PEL24–199, having reduced lysosomotropic activity, diminished the neurofibrillary degenerating process together with the short-term spatial memory in the Thy-Tau22 transgenic mouse model [23], and reduced amyloid burden in APPxPS1 transgenic mice model (to be published).

In the present study, a second family of compounds based on the same computer-assisted pharmacophoric design was synthesized, and 30 compounds around a diphenylpyrazole scaffold substituted with amino side chains were evaluated (D, Figure 2). One or two (dialkylamino)alkyl side chains were introduced at the 4-position of the pyrazole ring and/or the phenyl rings. Two series of compounds were then obtained with different (dialkylamino)alkyl or (heterocyclylamino)alkyl groups. The first series contained between two and four amino groups while the second had from one to four amino groups. Compounds were assumed to repress Aβ_1–x_ production, stabilize αCTF and AICD expression, and modulate autophagy through an increased expression of LC3-I/LC3-II and p62. Accordingly, all of these parameters as well as the pKa of compounds were therefore assessed. These parameters are shared by lysosomotropic drugs such as CQ, a weak di-base, or bafilomycin A1 (BafA1), which is an H^+^-ATPase proton-pump vacuolar inhibitor. Both CQ and bafilomycin have lysomotropic activity mediated by the alkalinization of cell-acidic compartments, leading to lysosome and autophagy-flux inhibition, and are, in addition, both toxic at µM concentrations [11,12]. APP metabolism is modified by this lysosomotropic activity with two consequences: (1) the repression of Aβ peptide production through indirect or non-competitive inhibition of BACE1, the β-secretase, and (2) the stabilization of both αCTF and AICD, where proteolysis occurs in lysosomes. Interestingly, in the previous series, PEL24–199 was selected because Aβ_1–x_ production was repressed without significant modifications in the CTFα or AICD quantities [22,23]. However, PEL24–199 was the sole compound showing both properties, suggesting that the non-competitive β-secretase inhibition differs from the lysosomotropic activities of CQ and BafA1 and that PEL24–199 activity occurs through a yet undetermined mechanism. Importantly, this unknown mechanism is unrelated to the alkalizing or lysosomotropic property of CQ or BafA1.

The lysosomotropic activity of CQ is mostly related to the internalization and accumulation of CQ in endolysosomal vesicles, in which this weak base alkalinizes the luminal content of endolysosomes. This intravesicular pH modification is then likely to be responsible for the non-competitive and indirect inhibition of β-secretase aspartyl protease activity since BACE1 activity is pH-dependent and optimal at acidic pH [32]. In the previous series of compounds, the loss of a unique basic group of the triamino compound PEL24–199 (X = CH instead of X = N, Figure 2) modified the pKas of the compound, possibly associated with the loss of the alkalizing property of the corresponding diamino compound. These compounds both enhanced αCTF and AICD expression to a greater extent than CQ [22]. In sharp contrast, all compounds of the present series had lower C_20_ than CQ, although this was non-related to their pKa and therefore, not related to the weak base property of these compounds. All pyrazole-derived compounds were more efficient to repress Aβ_1–x_ production with an IC_50_ inhibitory activity comprised between 1.2 and 12.4 µM and 1.7 to 13.2 µM for Aβ_1–40_ or Aβ_1–42_, respectively. Within these same cell-based assays, CQ has an IC_50_ of 7 and 12.7 µM for Aβ_1–40_ or Aβ_1–42_, respectively. Together, these results suggest that this inhibitory activity is different from that of CQ or even BafA1 [11,12], and is most likely not mediated by the weak base property. Moreover, several compounds including those having the greatest Aβ_1–x_ inhibitory activity also repressed the expression of sAPPβ, which is produced following APP cleavage by the β-secretase, suggesting that these pyrazole-derived compounds are also potent indirect or non-competitive inhibitors of β-secretase.

From the previous families of compounds, PEL24–199 [23] and RPEL [17] were shown to reduce both amyloid and neurofibrillary degeneration in vivo, suggesting that our compounds are also effective. Although modulation of the β-secretase activity has been suggested to also modulate Tau protein expression [33,34], the proteinaceous component of neurofibrillary degeneration, there is no direct evidence of the contribution of either BACE1 or BACE2 to Tau metabolism. Tau protein is a long half-life protein whose degradation is principally mediated by chaperone-mediated autophagy and more recently, intracellular Tau aggregate clearance was shown to occur through aggrephagy [35]. Compounds from previous families including PEL24–199 were shown to repress Tau aggregation and to modulate p62 and LC3-I/LC3-II, two master regulators of autophagy, also implicated in the aggrephagy. Increased expression of p62 and LC3-I/LC3-II following CQ or BafA1 treatment induced the accumulation of these proteins due to the blockade of the autophagy flux and lysosome activity. Herein, we showed that the inhibitory effect of our compounds toward Aβ_1–x_ expression was neither associated with p62 nor LC3-I/LC3-II. The non-competitive β-secretase effect is more related to the expression ratio of p62/(LCI3)/LC3II), in which individual expression is modified similarly between compounds of the present series. More precisely, p62 expression was less stimulated by compounds of family **A** than by compounds of family **B**. However, as for CQ and BafA1, the accumulation of p62 over 1.5-fold enhanced the cytotoxicity of the compounds similar to the loss of a protonable nitrogen in family **A**. Taking into account that all families are derived from a unique pharmacophore, itself derived from the structure–activity relationship of the most efficient compounds of family A [13,14] and family B [18,19], suggests a common mechanism for which the criteria of selection relies on (1) the repression of Aβ_1–x_ production without affecting the Aβ_x–38/40/42_ release; (2) a decrease ratio of p62/(LC3-I/LC3-II) expression below 0.7; and (3) an accumulation of p62 below 2.0-fold. The accumulation of APP metabolites αCTF and AICD by our compounds appears more likely to be unrelated to an alkalizing or lysosomotropic activity.

In conclusion, according to the present parameters of lead compound selection, compounds **24**, **65**, and **70** are those with comparable non-competitive β-secretase activity, low toxicity, and reduced lysosomotropic activity. Moreover, these common properties for several of our families of compounds suggest a similar mechanism of action that is likely to be mediated through a limited number, if not a single target interaction, and modulatory activity. However, target identification and the mechanism of action remain to be elucidated. This study is also bringing simple biochemical and biological parameters that could be useful for selecting drugs, which we showed to be active against both the pathophysiological processes of AD. The development of quantification methods of these parameters for high throughput screening would then enable the testing of already existing FDA-approved drugs and potentially help to decipher the mechanism of action.

## 4. Materials and Methods

### 4.1. Chemistry

All commercial reagents and solvents were used without further purification. Organic layers obtained after the extraction of aqueous solutions were dried over MgSO_4_ and filtered before evaporation. Reaction yields were not optimized. Column chromatography was performed using Macherey-Nagel silica gel (230–400 mesh). ^1^H and ^13^C NMR spectra were obtained using a Bruker DRX 300 spectrometer (Division BioSpin, Wissembourg, France) operating at 300.13 MHz for proton, operating at 300 MHz for ^1^H and 75 MHz for ^13^C, equipped with a BBFO 5 mm probe and a sample XpressLite. The data were processed using software TOPSPIN 4. Chemical shifts (δ) were expressed in ppm relative to either TMS or the residual proton signal in deuterated solvents. Mass spectra were recorded with an LC-MS (Waters Alliance Micromass ZQ 2000, Waters Corporation, Milford, MA, USA) using electrospray ionization. The purity of the final compounds was verified by two types of high-pressure liquid chromatography (HPLC) columns: C18 Interchrom UPTISPHERE and C4 Interchrom UPTISPHERE. Analytical HPLC was performed on a Shimadzu LC-2010AHT system equipped with a UV detector set at 254 nm and 215 nm. The following eluent systems were used: buffer A (H_2_O/TFA, 100:0.1) and buffer B (CH_3_CN/H_2_O/TFA, 80:20:0.1). Compounds were dissolved in 50 μL of buffer B and 950 μL of buffer A and injected into the system. HPLC retention times (HPLC *t_R_*) were obtained at a flow rate of 0.2 mL/min using a gradient run from 100% of buffer A to 100% of buffer B over 30 min. The spectra and chromatograms of final compounds can be found in Appendix A.

#### 4.1.1. General Procedure A

Dimethylformamide (81.5 mmol) was cooled to 0 °C with a salt/ice bath. Phosphorus oxychloride (22.5 mmol) was added dropwise with the temperature maintained below 0 °C. The mixture was then stirred for 40 min at 0 °C. Hydrazone (5.39 mmol) was added and the reaction mixture was allowed to warm to room temperature. After 2 h, the temperature was increased to 50 °C. The reaction was stirred at this temperature for 4 h. The mixture was then added to crushed ice and stirred for 1 h. Potassium carbonate was added until pH = 8 and the mixture was extracted twice with methylene chloride. The combined organic layers were washed with brine, dried, and evaporated. The residue was purified by column chromatography (DCM/MeOH/NH_4_OH = 9:1:0.1).

#### 4.1.2. General Procedure B

To a solution of alkylamine (3.73 mmol), 37% formaldehyde in water (22.4 mmol) and acetic acid (22.4 mmol) in methanol (20 mL) was slowly added sodium triacetoxyborohydride (18.7 mmol) over 30 min. The mixture was stirred until the completion of the reaction. The solution was an aqueous carbonate potassium (10%) and ethyl acetate was added to the residue and the mixture stirred for 10 min. The layers were separated, and the aqueous layer was extracted twice with ethyl acetate. The combined organic layers were washed with brine, dried, and evaporated. The residue was purified by column chromatography (DCM/MeOH-NH_3_ sat = 95:5).

#### 4.1.3. General Procedure C

LiAlH_4_ (1 M in THF, 2.82 mmol) was added to 20 mL of anhydrous THF under nitrogen. The solution was cooled to 0 °C with an ice bath and a solution of ester (1.88 mmol) in 20 mL of anhydrous THF was added dropwise. The reaction was stirred at 0 °C for 20 min and rt for 1 h. The mixture was then cooled with an ice bath and 0.11 mL of H_2_O was added, followed by 0.11 mL of 15% NaOH and 0.33 mL of H_2_O. The solid was isolated by filtration and washed with THF. The filtrate was evaporated. The residue was purified by column chromatography (DCM/MeOH/NH_4_OH = 9:1:0.1).

#### 4.1.4. General Procedure D

To a solution of the desired compound (1.42 mmol) in chloroform (25 mL) was added manganese(IV) oxide (14.2 mmol). This was stirred at rt for 24 h. More manganese(IV) oxide (14.2 mmol) was added, and the mixture was stirred for 24 h. The solid was filtered off and washed with methylene chloride. The filtrate was washed with 10% K_2_CO_3_ and with brine. The organic layer was dried and evaporated to give the aldehyde that was used in the next step without further purification.

#### 4.1.5. General Procedure E

To a solution of benzaldehyde (0.265 mmol), amine (0.45 mmol), and acetic acid (0.53 mmol) in DCE (4 mL) was added sodium triacetoxyborohydride (0.53 mmol). The reaction mixture was stirred under nitrogen for 24 h. A total of 10% K_2_CO_3_ was added, and the layers were separated. The aqueous layer was extracted twice with methylene chloride. The combined organic layers were washed with brine, dried, and evaporated. The residue was purified by column chromatography (DCM/MeOH-NH_3_ sat = 9:1).

#### 4.1.6. Methyl 4-Hydrazinobenzoate Hydrochloride (**3**)

A solution of **1** (6 g, 39.7 mmol) in HCl 37% (40 mL) was brought to −5 °C with a salt/ice bath. A solution of sodium nitrite (3 g, 43.5 mmol) in water (22 mL) was slowly added over 1 h while maintaining the temperature below 0 °C. The solution was then stirred at 0 °C for 40 min and a solution of tin(II) chloride (13.86 g, 48.7 mmol) in 37% HCl (20 mL) was added dropwise while maintained at 0 °C. The mixture was stirred for another 20 min at 0 °C and 2 h 30 min at room temperature. The precipitate was collected by filtration, washed with 40 mL of ice-cold water, and dried to give 8.6 g of a white solid, which was used for the next step without further purification.

#### 4.1.7. Methyl 3-Hydrazinobenzoate Hydrochloride (**4**)

A solution of **2** (4 g, 26.5 mmol) in 37% HCl (40 mL) was brought to −5 °C with a salt/ice bath. A solution of sodium nitrite (2 g, 29 mmol) in water (15 mL) was slowly added over 1 h while maintaining the temperature below 3 °C. The solution was then stirred at 0 °C for 30 min and a solution of tin(II) chloride (9.24 g, 48.7 mmol) in 37% HCl (20 mL) was added dropwise while maintaining the temperature at 0 °C. The mixture was stirred for another 30 min at 0 °C and 2 h at room temperature. The precipitate was collected by filtration, washed subsequently with 15 mL of ice-cold water and with ether, and dried to give 6.55 g of a white solid, which was used for the next step without further purification.

#### 4.1.8. 1-[4-[3-(Dimethylamino)propyl]phenyl]ethanone (**5**)

In a sealed tube, a mixture of 1-[4-(3-bromopropyl)phenyl]ethanone (5.6 g, 23.2 mmol) and dimethylamine (2 M in methanol, 34.8 mL, 69.6 mmol) was heated at 65 °C for 15 h. The solvent was evaporated and 10% K_2_CO_3_ (100 mL) was added. The mixture was extracted twice with ethyl acetate and the combined organic layers were washed with brine, dried, and evaporated. The residue was purified by column chromatography (DCM/MeOH = 9:1) to give 3.66 g (59%) of the product as a yellow oil. ^1^H NMR (CDCl_3_, 300 Mz): *δ* 7.89 (d, *J* = 8.4 Hz, 2H), 7.29 (d, *J* = 8.4 Hz, 2H), 2.70 (t, *J* = 7.6 Hz, 2H), 2.58 (s, 3H), 2.29 (t, *J* = 7.1 Hz, 2H), 2.23 (s, 6H), 1.81 (quint, *J* = 7.5 Hz, 2H). ^13^C NMR (CDCl_3_, 75 MHz): *δ* 197.9, 148.2, 135.1, 128.7, 128.6, 59.0, 45.5, 33.7, 29.1, 26.6. MS (ESI) *m*/*z* 206 [M + H]^+^.

#### 4.1.9. Methyl 4-[2-[1-[4-[3-(Dimethylamino)ropyl]phenyl]ethylidene]hydrazino] Benzoate (**6**)

A solution of **3** (8.5 g) in methanol (220 mL) was added to **5** (3.45 g, 16.81 mmol). The mixture was stirred at rt for 48 h. The precipitate was collected by filtration and washed with methanol to give 4.08 g (62%) of the product as a yellowish solid. ^1^H NMR (CD_3_SOCD_3_, 300 Mz): *δ* 10.88 (br, 1H), 9.89 (s, 1H), 7.83 (d, *J* = 8.8 Hz, 1H), 7.75 (d, *J* = 8.3 Hz, 1H), 7.33 (d, *J* = 8.8 Hz, 1H), 7.27 (d, *J* = 8.3 Hz, 1H), 3.78 (s, 3H), 3.02–2.99 (m, 2H), 2.71–2.63 (m, 8H), 2.30 (s, 3H), 2.05–1.95 (m, 2H). ^13^C NMR (CD_3_SOCD_3_, 75 MHz): *δ* 166.2, 150.0, 143.5, 140.5, 136.8, 130.8, 128.3, 125.6, 119.1, 112.0, 56.0, 51.4, 41.9, 31.6, 25.1, 13.3. MS (ESI) *m*/*z* 354 [M + H]^+^.

#### 4.1.10. Methyl 3-[2-[1-[4-[3-(Dimethylamino)propyl]phenyl]ethylidene]hydrazino] Benzoate Hydrochloride (**7**)

A solution of **4** (10.9 g) in methanol (105 mL) was added to **5** (3.45 g, 16.81 mmol). The mixture was stirred at rt for 48 h. The precipitate was collected by filtration and washed with methanol to give 6.19 g (94%) of the product as a light brown solid. ^1^H NMR (CD_3_SOCD_3_, 300 Mz): *δ* 10.30 (br, 1H), 9.52 (s, 1H), 7.84–7.83 (m, 1H), 7.73 (d, *J* = 8.2 Hz, 2H), 7.53–7.49 (m, 1H), 7.36–7.34 (m, 2H), 7.26 (d, *J* = 8.3 Hz, 2H), 3.85 (s, 3H), 3.05–3.00 (m, 2H), 2.73 (s, 6H), 2.65 (t, *J* = 7.7 Hz, 2H), 2.26 (s, 3H), 2.03–1.93 (m, 2H). ^13^C NMR (CD_3_SOCD_3_, 75 MHz): *δ* 166.6, 146.4, 141.9, 140.1, 137.1, 130.3, 129.3, 128.3, 125.4, 119.3, 117.1, 113.3, 56.1, 52.0, 42.0, 31.6, 25.2, 13.1. MS (ESI) *m*/*z* 354 [M + H]^+^.

#### 4.1.11. Methyl 4-[3-[4-[3-(Dimethylamino)propyl]phenyl]-4-formyl-pyrazol-1-yl] Benzoate (**8**)

General procedure A: 87% yield (white solid). ^1^H NMR (CDCl_3_, 300 Mz): *δ* 10.03 (s, 1H), 8.59 (s, 1H), 8.14 (d, *J* = 8.8 Hz, 2H), 7.86 (d, *J* = 8.8 Hz, 2H), 7.72 (d, *J* = 8.1 Hz, 2H), 7.31 (d, *J* = 8.1 Hz, 2H), 3.92 (s, 3H), 2.70 (t, *J* = 7.5 Hz, 2H), 2.32 (t, *J* = 7.1 Hz, 2H), 2.23 (s, 6H), 1.82 (quint, *J* = 7.4 Hz, 2H). ^13^C NMR (CDCl_3_, 75 MHz): *δ* 185.1, 166.0, 155.2, 144.0, 142.1, 131.3, 131.2, 129.3, 128.9, 128.6, 123.0, 119.0, 59.1, 52.4, 45.5, 33.5, 29.3. MS (ESI) *m*/*z* 392 [M + H]^+^.

#### 4.1.12. Methyl 3-[3-[4-[3-(Dimethylamino)propyl]phenyl]-4-formyl-pyrazol-1-yl] Benzoate (**9**)

General procedure A: 67% yield (white solid). ^1^H NMR (CDCl_3_, 300 Mz): *δ* 10.06 (s, 1H), 8.61 (s, 1H), 8.44–8.42 (m, 1H), 8.05–8.02 (m, 1H), 7.76 (d, *J* = 8.1 Hz, 2H), 7.58 (t, *J* = 7.9 Hz, 1H), 7.33 (d, *J* = 8.1 Hz, 2H), 3.96 (s, 3H), 2.72 (t, *J* = 7.5 Hz, 2H), 2.33 (t, *J* = 7.1 Hz, 2H), 2.23 (s, 6H), 1.85 (quint, *J* = 7.3 Hz, 2H). ^13^C NMR (CDCl_3_, 75 MHz): *δ* 185.1, 165.9, 154.9, 143.9, 139.2, 131.8, 131.2, 129.9, 128.9, 128.9, 128.7, 128.7, 123.8, 122.8, 120.3, 59.2, 52.5, 45.5, 33.5, 29.3. MS (ESI) *m*/*z* 392 [M + H]^+^.

#### 4.1.13. Methyl 4-[4-(Dimethylaminomethyl)-3-[4-[3-(dimethylamino)propyl]phenyl] Pyrazol-1-yl]benzoate (**10**)

A mixture of **8** (0.9 g, 2.3 mmol), dimethylamine (2 M in THF, 2.3 mL, 4.6 mmol), sodium triacetoxyborohydride (0.88 g, 4.14 mmol), and acetic acid (0.24 mL, 4.14 mmol) in DCE (10 mL) was stirred at rt under nitrogen for 4 h. A total of 10% K_2_CO_3_ and methylene chloride were added. The layers were separated, and the aqueous layer was extracted twice with methylene chloride. The combined organic layers were washed with brine, dried, and evaporated. The residue was purified by column chromatography (DCM/MeOH/NH_4_OH = 9:1:0.05) to give 0.82 g (84%) of the product as a white solid. ^1^H NMR (CD_3_OD, 300 Mz): *δ* 8.22 (s, 1H), 8.03 (d, *J* = 8.8 Hz, 2H), 7.84 (d, *J* = 8.8 Hz, 2H), 7.72 (d, *J* = 8.1 Hz, 2H), 7.26 (d, *J* = 8.1 Hz, 2H), 3.86 (s, 3H), 3.43 (s, 2H), 2.64 (t, *J* = 7.6 Hz, 2H), 2.37–2.32 (m, 2H), 2.22 (s, 6H), 2.20 (s, 6H), 1.86–1.79 (m, 2H). ^13^C NMR (CD_3_OD, 75 MHz): *δ* 167.6, 154.5, 144.4, 143.3, 132.0, 131.9, 130.3, 129.5 (2C), 128.5, 119.8, 118.9, 60.1, 54.1, 52.6, 45.4, 45.2, 34.3, 29.9. MS (ESI) *m*/*z* 421 [M + H]^+^.

#### 4.1.14. Methyl 3-[4-(Dimethylaminomethyl)-3-[4-[3-(dimethylamino)propyl]phenyl] Pyrazol-1-yl]benzoate (**11**)

To a solution of **9** (0.7 g, 1.79 mmol), dimethylamine (2 M in THF, 1.79 mL, 3.58 mmol) and acetic acid (0.18 mL, 3.22 mmol) in DCE (8 mL) was added sodium triacetoxyborohydride (0.68 g, 3.22 mmol). The reaction mixture was stirred at rt for 4 h. A total of 10% K_2_CO_3_ was added, and the layers were separated. The organic layer was extracted twice with methylene chloride. The combined organic layers were washed with brine, dried, and evaporated. The residue was purified by column chromatography (DCM/MeOH = 95:5 to 8:2) to give 0.64 g (85%) of the product as a colorless oil. ^1^H NMR (CD_3_OD, 300 Mz): *δ* 8.37 (s, 1H), 8.19 (s, 1H), 7.97–7.94 (m, 1H), 7.84 (d, *J* = 7.9 Hz, 1H), 7.71 (d, *J* = 8.1 Hz, 2H), 7.49 (t, *J* = 7.9 Hz, 1H), 7.25 (d, *J* = 8.1 Hz, 2H), 3.88 (s, 3H), 3.44 (s, 2H), 2.62 (t, *J* = 7.6 Hz, 2H), 2.36 (t, *J* = 7.4 Hz, 2H), 2.24 (s, 6H), 2.20 (s, 6H), 1.81 (quint, *J* = 7.8 Hz, 2H). ^13^C NMR (CD_3_OD, 75 MHz): *δ* 167.4, 154.0, 143.0, 141.3, 132.6, 132.0, 130.8, 130.1, 129.5 (2C), 127.9, 123.7, 120.2, 119.2, 59.9, 54.0, 52.8, 45.3, 45.2, 34.2, 29.7. MS (ESI) *m*/*z* 421 [M + H]^+^.

#### 4.1.15. Methyl 4-[4-(3-Aminopropyl)-3-[4-[3-(dimethylamino)propyl]phenyl] Pyrazol-1-yl]benzoate (**12**)

To a suspension of NaH (60% dispersion in mineral oil, 0.17 g, 4.32 mmol) in anhydrous THF (15 mL) at 0 °C under nitrogen was added dropwise diethyl cyanomethylphosphonate (0.65 mL, 3.99 mmol). The mixture was stirred at 0 °C for 30 min and a solution of **8** (1.3 g, 3.32 mmol) in anhydrous THF (20 mL) was added dropwise. The reaction mixture was stirred at 0 °C for 10 min and then allowed to warm to rt. After 2 h, the solvent was evaporated, and the residue was purified by column chromatography (DCM/MeOH/NH_4_OH = 9:1:0.02) to give 1.175 g (85%) of a white solid. MS (ESI) *m*/*z* 415 [M + H]^+^.

A mixture of the intermediate (1.16 g, 2.80 mmol), Raney Nickel (0.12 g), and 10% Pd/C (0.12 g) in methanol saturated with ammonia (110 mL) and THF (10 mL) was stirred under a hydrogen atmosphere for 30 h. The catalyst was filtered off and the filtrate was evaporated. The residue was purified by column chromatography (DCM/MeOH/NH_4_OH = 9:1:0.1) to give 0.795 g (68%) of the product as a colorless oil. ^1^H NMR (CDCl_3_, 300 Mz): *δ* 8.12 (d, *J* = 8.9 Hz, 2H), 7.88 (s, 1H), 7.82 (d, *J* = 8.9 Hz, 2H), 7.65 (d, *J* = 8.1 Hz, 2H), 7.28 (d, *J* = 8.1 Hz, 2H), 3.93 (s, 3H), 2.81–2.67 (m, 6H), 2.37–2.32 (m, 2H), 2.26 (s, 6H), 2.00 (br, 2H), 1.90–1.76 (m, 4H). ^13^C NMR (CDCl_3_, 75 MHz): *δ* 166.6, 152.6, 143.4, 142.3, 131.2, 131.0, 128.8, 128.0, 127.3, 126.2, 122.2, 117.7, 59.3, 52.3, 45.5, 41.8, 33.8, 33.5, 29.3, 22.1. MS (ESI) *m*/*z* 421 [M + H]^+^.

#### 4.1.16. Methyl 3-[4-(3-Aminopropyl)-3-[4-[3-(dimethylamino)propyl]phenyl] Pyrazol-1-yl]benzoate (**13**)

To a suspension of NaH (60% dispersion in mineral oil, 0.1 g, 2.5 mmol) in 12 mL of anhydrous THF at 0 °C under nitrogen was added dropwise a solution of diethyl cyanomethylphosphonate (0.38 mL, 2.32 mmol) in anhydrous THF (3 mL). The mixture was stirred at 0 °C for 30 min and **9** (0.7 g, 1.79 mmol) was slowly added. The reaction mixture was allowed to warm to room temperature. After 2 h, water and ethyl acetate were added, and the layers were separated. The aqueous layer was extracted with ethyl acetate. The combined organic layers were washed with brine, dried, and evaporated. The residue was purified by column chromatography (DCM/MeOH-NH_3_ sat = 9:1) to give 527 mg (71%) of a colorless oil. MS (ESI) *m*/*z* 415 [M + H]^+^.

A mixture of the intermediate (0.5 g, 1.21 mmol), Raney Nickel (50 mg), and 10% Pd/C (50 mg) in methanol saturated with ammonia (60 mL) was stirred under a hydrogen atmosphere for 30 h. The catalyst was filtered off and the filtrate was evaporated. The residue was purified by column chromatography (DCM/MeOH/NH_4_OH = 9:1:0.1) to give 317 mg (62%) of the product as a colorless oil. ^1^H NMR (CDCl_3_, 300 Mz): *δ* 8.28–8.27 (m, 1H), 7.96–7.92 (m, 1H), 7.86–7.82 (m, 1H), 7.81 (s, 1H), 7.60 (d, *J* = 8.1 Hz, 2H), 7.43 (t, *J* = 7.9 Hz, 1H), 7.22 (d, *J* = 8.1 Hz, 2H), 3.88 (s, 3H), 2.73–2.60 (m, 6H), 2.26 (t, *J* = 7.1 Hz, 2H), 2.18 (s, 6H), 1.90 (br, 2H), 1.82–1.68 (m, 4H). ^13^C NMR (CDCl_3_, 75 MHz): *δ* 166.4, 151.8, 142.0, 140.1, 131.3, 131.0, 129.5, 128.5, 127.7, 126.6, 125.9, 122.7, 121.5, 118.9, 59.1, 52.2, 45.4, 41.6, 33.9, 33.3, 29.2, 22.0. MS (ESI) *m*/*z* 421 [M + H]^+^.

#### 4.1.17. Methyl 4-[4-[3-(Dimethylamino)propyl]-3-[4-[3-(dimethylamino) Propyl]phenyl]pyrazol-1-yl]benzoate (**14**)

General procedure B: 90% yield (white solid). ^1^H NMR (CDCl_3_, 300 Mz): *δ* 8.11 (d, *J* = 8.8 Hz, 2H), 7.88 (s, 1H), 7.82 (d, *J* = 8.8 Hz, 2H), 7.65 (d, *J* = 8.1 Hz, 2H), 7.27 (d, *J* = 8.1 Hz, 2H), 3.92 (s, 3H), 2.74–2.66 (m, 4H), 2.37–2.32 (m, 4H), 2.26 (s, 6H), 2.23 (s, 6H), 1.89–1.77 (m, 4H). ^13^C NMR (CDCl_3_, 75 MHz): *δ* 166.6, 152.6, 143.4, 142.2, 131.2, 131.0, 128.7, 128.0, 127.2, 126.2, 122.3, 117.7, 59.3, 59.2, 52.3, 45.5, 45.5, 33.5, 29.3, 28.2, 22.6. MS (ESI) *m*/*z* 449 [M + H]^+^.

#### 4.1.18. Methyl 3-[4-[3-(Dimethylamino)propyl]-3-[4-[3-(dimethylamino)propyl] Phenyl]pyrazol-1-yl]benzoate (**15**)

General procedure B: 82% yield (colorless oil). ^1^H NMR (CD_3_OD, 300 Mz): *δ* 8.37–8.36 (m, 1H), 8.10 (s, 1H), 7.98–7.94 (m, 1H), 7.85–7.83 (m, 1H), 7.61 (d, *J* = 8.1 Hz, 2H), 7.50 (t, *J* = 8.0 Hz, 1H), 7.26 (d, *J* = 8.1 Hz, 2H), 3.89 (s, 3H), 2.66–2.61 (m, 4H), 2.39–2.31 (m, 4H), 2.24 (s, 6H), 2.20 (s, 6H), 1.87–1.74 (m, 4H). ^13^C NMR (CD_3_OD, 75 MHz): *δ* 167.6, 153.0, 143.0, 141.5, 132.6, 132.5, 130.8, 129.6, 129.0, 128.2, 127.7, 123.6, 122.7, 120.1, 60.1, 60.0, 52.9, 45.3 (2C), 34.3, 29.9, 28.6, 23.5. MS (ESI) *m*/*z* 449 [M + H]^+^.

#### 4.1.19. [4-[4-(Dimethylaminomethyl)-3-[4-[3-(dimethylamino)propyl]phenyl] Pyrazol-1-yl]phenyl]methanol (**16**)

General procedure C: 88% yield (colorless oil). ^1^H NMR (CD_3_OD, 300 Mz): *δ* 8.18 (s, 1H), 7.76 (d, *J* = 8.6 Hz, 2H), 7.68 (d, *J* = 8.2 Hz, 2H), 7.46 (d, *J* = 8.6 Hz, 2H), 7.29 (d, *J* = 8.1 Hz, 2H), 4.63 (s, 2H), 3.50 (s, 2H), 2.66 (t, *J* = 7.6 Hz, 2H), 2.39–2.34 (m, 2H), 2.24 (s, 6H), 2.22 (s, 6H), 1.89–1.78 (m, 2H). ^13^C NMR (CD_3_OD, 75 MHz): *δ* 153.9, 143.2, 141.3, 140.3, 132.1, 130.4, 129.6, 129.6, 129.1, 119.9, 118.6, 64.5, 60.1, 53.9, 45.4, 45.1, 34.3, 30.0. MS (ESI) *m*/*z* 393 [M + H]^+^.

#### 4.1.20. [4-[4-[3-(Dimethylamino)propyl]-3-[4-[3-(dimethylamino)propyl]phenyl] Pyrazol-1-yl]phenyl]methanol (**17**)

General procedure C: 89% yield (white solid). ^1^H NMR (CD_3_OD, 300 Mz): *δ* 8.08 (s, 1H), 7.74 (d, *J* = 8.6 Hz, 2H), 7.61 (d, *J* = 8.2 Hz, 2H), 7.44 (d, *J* = 8.7 Hz, 2H), 7.29 (d, *J* = 8.2 Hz, 2H), 4.62 (s, 2H), 2.69–2.64 (m, 4H), 2.38–2.30 (m, 4H), 2.23 (s, 6H), 2.19 (s, 6H), 1.88–1.72 (m, 4H). ^13^C NMR (CD_3_OD, 75 MHz): *δ* 152.7, 143.0, 141.0, 140.4, 132.6, 129.6, 129.1, 129.1, 128.4, 122.2, 119.7, 64.6, 60.2, 60.1, 45.4 (2C), 34.3, 30.0, 28.8, 23.4. MS (ESI) *m*/*z* 421 [M + H]^+^.

#### 4.1.21. [3-[4-(Dimethylaminomethyl)-3-[4-[3-(dimethylamino)propyl]phenyl]pyrazol-1-yl]phenyl]methanol (**18**)

General procedure C: 92% yield (colorless oil). ^1^H NMR (CDCl_3_, 300 Mz): *δ* 7.94 (s, 1H), 7.76–7.73 (m, 3H), 7.59 (d, *J* = 8.1 Hz, 1H), 7.31 (t, *J* = 7.7 Hz, 1H), 7.23–7.16 (m, 3H), 5.89 (s, 1H), 4.63 (s, 2H), 3.42 (s, 2H), 2.61 (t, *J* = 7.5 Hz, 2H), 2.28 (t, *J* = 7.0 Hz, 2H), 2.22 (s, 6H), 2.18 (s, 6H), 1.79 (quint, *J* = 7.3 Hz, 2H). ^13^C NMR (CDCl_3_, 75 MHz): *δ* 151.8, 143.5, 141.3, 139.7, 130.7, 129.0, 128.2, 128.0, 127.8, 123.9, 117.8, 116.9, 116.4, 63.6, 58.7, 53.4, 44.9, 44.8, 33.1, 28.6. MS (ESI) *m*/*z* 393 [M + H]^+^.

#### 4.1.22. [3-[4-[3-(Dimethylamino)propyl]-3-[4-[3-(dimethylamino)propyl]phenyl]Pyrazol-1-yl]phenyl]methanol (**19**)

General procedure C: 88% yield (colorless oil). ^1^H NMR (CD_3_OD, 300 Mz): *δ* 8.08 (s, 1H), 7.79 (s, 1H), 7.67–7.60 (m, 3H), 7.42 (t, *J* = 7.7 Hz, 1H), 7.29–7.27 (m, 3H), 4.67 (s, 2H), 2.69–2.63 (m, 4H), 2.38–2.30 (m, 4H), 2.23 (s, 6H), 2.19 (s, 6H), 1.88–1.72 (m, 4H). ^13^C NMR (CD_3_OD, 75 MHz): *δ* 152.8, 144.7, 143.1, 141.4, 132.6, 130.5, 129.6, 129.1, 128.5, 125.6, 122.3, 118.6, 118.1, 64.8, 60.2, 60.1, 45.4 (2C), 34.3, 30.0, 28.8, 23.4. MS (ESI) *m*/*z* 421 [M + H]^+^.

#### 4.1.23. 4-[4-(Dimethylaminomethyl)-3-[4-[3-(dimethylamino)propyl]phenyl] Pyrazol-1-yl] benzaldehyde (**20**)

General procedure D: 72% yield (colorless oil). ^1^H NMR (CDCl_3_, 300 Mz): *δ* 9.96 (s, 1H), 8.01 (s, 1H), 7.95–7.89 (m, 4H), 7.78 (d, *J* = 8.2 Hz, 2H), 7.26 (d, *J* = 8.2 Hz, 2H), 3.44 (s, 2H), 2.67 (t, *J* = 7.6 Hz, 2H), 2.32–2.27 (m, 8H), 2.21 (s, 6H), 1.86–1.76 (m, 2H). ^13^C NMR (CDCl_3_, 75 MHz): *δ* 191.0, 153.5, 144.2, 142.5, 133.7, 131.3, 130.4, 128.6, 128.3, 128.0, 120.3, 118.2, 59.2, 53.9, 45.5, 45.3, 33.5, 29.3. MS (ESI) *m*/*z* 391 [M + H]^+^.

#### 4.1.24. 4-[4-[3-(Dimethylamino)propyl]-3-[4-[3-(dimethylamino)propyl]phenyl] Pyrazol-1-yl]benzaldehyde (**21**)

General procedure D: 76% yield (yellow oil). ^1^H NMR (50 °C, CD_3_OD, 300 Mz): *δ* 9.92 (s, 1H), 8.16 (s, 1H), 7.95 (s, 4H), 7.64–7.61 (m, 2H), 7.29–7.27 (m, 2H), 2.70–2.64 (m, 4H), 2.39–2.31 (m, 4H), 2.24 (s, 6H), 2.20 (s, 6H), 1.86–1.75 (m, 4H). ^13^C NMR (50°C, CD_3_OD, 75 MHz): *δ* 192.7, 154.2, 145.6, 143.4, 135.3, 132.3, 132.2, 129.6, 129.1, 128.5, 123.7, 119.3, 60.2, 60.2, 45.4 (2C), 34.4, 29.8, 28.7, 23.5. MS (ESI) *m*/*z* 419 [M + H]^+^.

#### 4.1.25. 3-[4-(Dimethylaminomethyl)-3-[4-[3-(dimethylamino)propyl]phenyl] Pyrazol-1-yl] Benzaldehyde (**22**)

General procedure D: 60% yield (yellow oil). ^1^H NMR (CDCl_3_, 300 Mz): *δ* 10.04 (s, 1H), 8.24–8.23 (m, 1H), 8.08–8.04 (m, 1H), 8.01 (s, 1H), 7.82 (d, *J* = 8.1 Hz, 2H), 7.72 (d, *J* = 7.6 Hz, 1H), 7.58 (t, *J* = 7.8 Hz, 1H), 7.29 (d, *J* = 8.2 Hz, 2H), 3.45 (s, 2H), 2.69 (t, *J* = 7.6 Hz, 2H), 2.35–2.29 (m, 8H), 2.24 (s, 6H), 1.84 (quint, *J* = 7.4 Hz, 2H). ^13^C NMR (CD_3_OD, 75 MHz): *δ* 193.2, 154.3, 143.3, 141.9, 139.1, 132.0, 130.3, 129.6, 129.5, 128.2, 125.0, 119.9, 119.4, 118.7, 60.1, 54.0, 45.4, 45.2, 34.3, 29.9. MS (ESI) *m*/*z* 391 [M + H]^+^.

#### 4.1.26. 3-[4-[3-(Dimethylamino)propyl]-3-[4-[3-(dimethylamino)propyl]phenyl] Pyrazol-1-yl]benzaldehyde (**23**)

General procedure D: 66% yield (colorless oil). ^1^H NMR (CD_3_OD, 50 °C, 300 Mz): *δ* 10.03 (s, 1H), 8.28–8.27 (m, 1H), 8.16 (s, 1H), 8.09–8.04 (m, 1H), 7.79–7.76 (m, 1H), 7.66–7.59 (m, 3H), 7.31–7.27 (m, 2H), 2.73–2.66 (m, 4H), 2.49–2.40 (m, 4H), 2.31 (s, 6H), 2.25 (s, 6H), 1.89–1.78 (m, 4H). ^13^C NMR (CD_3_OD, 50 °C, 75 MHz): *δ* 193.3, 153.5, 143.1, 142.1, 139.3, 132.5, 131.4, 129.6, 129.2, 128.4, 128.0, 125.1, 122.9, 119.8, 60.0, 60.0, 45.2, 45.2, 34.2, 29.5, 28.5, 23.3. MS (ESI) *m*/*z* 419 [M + H]^+^.

#### 4.1.27. 3-[4-[4-(Dimethylaminomethyl)-1-[4-(dimethylaminomethyl)phenyl] Pyrazol-3-yl]phenyl]-*N*,*N*-dimethyl-propan-1-amine (**24**)

General procedure E: 80% yield. The compound was converted to its 3HCl salt (white solid). ^1^H NMR (free amine, CD_3_OD, 300 Mz): *δ* 8.21 (s, 1H), 7.77 (d, *J* = 8.6 Hz, 2H), 7.70 (d, *J* = 8.2 Hz, 2H), 7.43 (d, *J* = 8.6 Hz, 2H), 7.30 (t, *J* = 8.2 Hz, 2H), 3.52 (s, 2H), 3.49 (s, 2H), 2.68 (t, *J* = 7.6 Hz, 2H), 2.40–2.35 (m, 2H), 2.25 (s, 12H), 2.23 (s, 6H), 1.87–1.82 (m, 2H). ^13^C NMR (free amine, CD_3_OD, 75 MHz): *δ* 154.0, 143.2, 140.5, 137.2, 132.2, 131.8, 130.4, 129.6, 129.6, 119.8, 118.8, 64.2, 60.1, 54.0, 45.4, 45.2, 45.1, 34.3, 30.0. MS (ESI) *m*/*z* 420 [M + H]^+^. P_HPLC_ > 97%. HPLC (C_4_, 35 min): *t_R_* 8.4 min, P_HPLC_ 99%; HPLC (C_18_, 35 min): *t_R_* 10.8 min, P_HPLC_ 97%.

#### 4.1.28. 3-[4-[1-[4-(Dimethylaminomethyl)phenyl]-4-[3-(dimethylamino)propyl] Pyrazol-3-yl]phenyl]-*N*,*N*-dimethyl-propan-1-amine (**25**)

General procedure E: 89% yield. The compound was converted to its 3HCl salt (white solid). ^1^H NMR (free amine, CD_3_OD, 300 Mz): *δ* 8.13 (s, 1H), 7.76 (d, *J* = 8.5 Hz, 2H), 7.63 (d, *J* = 8.1 Hz, 2H), 7.41 (d, *J* = 8.5 Hz, 2H), 7.30 (d, *J* = 8.1 Hz, 2H), 3.50 (s, 2H), 2.72–2.65 (m, 4H), 2.48–2.40 (m, 4H), 2.31 (s, 6H), 2.26 (s, 6H), 2.25 (s, 6H), 1.92–1.76 (m, 4H). ^13^C NMR (free amine, CD_3_OD, 75 MHz): *δ* 152.7, 142.8, 140.7, 136.7, 132.6, 131.8, 129.6, 129.1, 128.4, 122.1, 119.6, 64.1, 60.0, 59.9, 45.2 (3C), 34.2, 29.7, 28.5, 23.3. MS (ESI) *m*/*z* 448 [M + H]^+^. P_HPLC_ > 96%. HPLC (C_4_, 35 min): *t_R_* 8.7 min, P_HPLC_ 99%; HPLC (C_18_, 35 min): *t_R_* 11.8 min, P_HPLC_ 96%.

#### 4.1.29. 3-[4-[4-(Dimethylaminomethyl)-1-[3-(dimethylaminomethyl)phenyl] Pyrazol-3-yl]phenyl]-*N*,*N*-dimethyl-propan-1-amine (**26**)

General procedure E: 72% yield. The compound was converted to its 3HCl salt (white solid). ^1^H NMR (free amine, CDCl_3_, 300 Mz): *δ* 7.97 (s, 1H), 7.79 (d, *J* = 8.2 Hz, 2H), 7.71 (s, 1H), 7.67 (d, *J* = 8.0 Hz, 1H), 7.37 (t, *J* = 7.7 Hz, 1H), 7.25 (d, *J* = 8.3 Hz, 2H), 7.20 (d, *J* = 7.6 Hz, 1H), 3.46 (s, 2H), 3.46 (s, 2H), 2.68 (t, *J* = 7.6 Hz, 2H), 2.37 (t, *J* = 7.2 Hz, 2H), 2.28 (s, 6H), 2.27 (s, 6H), 2.25 (s, 6H), 1.86 (quint, *J* = 7.4 Hz, 2H). ^13^C NMR (CDCl_3_, 75 MHz): *δ* 152.2, 141.6, 140.6, 140.1, 131.1, 129.3, 128.5, 128.3, 128.1, 126.7, 119.1, 118.4, 117.6, 64.2, 59.1, 53.9, 45.5, 45.2, 45.2, 33.4, 29.0. MS (ESI) *m*/*z* 420 [M + H]^+^. P_HPLC_ > 97%. HPLC (C_4_, 35 min): *t_R_* 7.6 min, P_HPLC_ 98%; HPLC (C_18_, 35 min): *t_R_* 11.1 min, P_HPLC_ 97%.

#### 4.1.30. 3-[4-[1-[3-(Dimethylaminomethyl)phenyl]-4-[3-(dimethylamino)propyl] Pyrazol-3-yl]phenyl]-*N*,*N*-dimethyl-propan-1-amine (**27**)

General procedure E: 68% yield. The compound was converted to its 3HCl salt (white solid). ^1^H NMR (free amine, CD_3_OD, 300 Mz): *δ* 8.12 (s, 1H), 7.77 (s, 1H), 7.73–7.69 (m, 1H), 7.63 (d, *J* = 8.2 Hz, 2H), 7.43 (t, *J* = 7.8 Hz, 1H), 7.29 (d, *J* = 8.1 Hz, 2H), 7.23 (d, *J* = 7.6 Hz, 1H), 3.52 (s, 2H), 2.72–2.65 (m, 4H), 2.42–2.34 (m, 4H), 2.26 (s, 12H), 2.22 (s, 6H), 1.90–1.77 (m, 4H). ^13^C NMR (free amine, CD_3_OD, 75 MHz): *δ* 152.8, 143.0, 141.4, 140.7, 132.6, 130.5, 129.6, 129.1, 128.4, 128.3, 122.3, 120.7, 118.8, 64.7, 60.2, 60.1, 45.3 (3C), 34.3, 29.9, 28.8, 23.4. MS (ESI) *m*/*z* 448 [M + H]^+^. P_HPLC_ > 98%. HPLC (C_4_, 35 min): *t_R_* 8.9 min, P_HPLC_ 98%; HPLC (C_18_, 35 min): *t_R_* 12.1 min, P_HPLC_ 98%.

#### 4.1.31. 3-[4-[4-(Dimethylaminomethyl)-1-[4-[(4-methylpiperazin-1-yl)methyl] Phenyl]pyrazol-3-yl]phenyl]-*N*,*N*-dimethyl-propan-1-amine (**28**)

General procedure E: 75% yield. The compound was converted to its 4HCl salt (white solid). ^1^H NMR (free amine, CD_3_OD, 300 Mz): *δ* 8.21 (s, 1H), 7.76 (d, *J* = 8.6 Hz, 2H), 7.70 (d, *J* = 8.2 Hz, 2H), 7.44 (d, *J* = 8.6 Hz, 2H), 7.30 (t, *J* = 8.2 Hz, 2H), 3.55 (s, 2H), 3.52 (s, 2H), 2.68 (t, *J* = 7.6 Hz, 2H), 2.49 (br, 8H), 2.42–2.37 (m, 2H), 2.26 (s, 9H), 2.24 (s, 6H), 1.90–1.80 (m, 2H). ^13^C NMR (free amine, CD_3_OD, 75 MHz): *δ* 153.9, 143.2, 140.4, 137.1, 132.2, 131.7, 130.4, 129.6, 129.6, 119.8, 118.7, 63.1, 60.1, 55.7, 54.0, 53.5, 46.0, 45.3, 45.1, 34.3, 29.9. MS (ESI) *m*/*z* 475 [M + H]^+^. P_HPLC_ > 97%. HPLC (C_4_, 35 min): *t_R_* 7.6 min, P_HPLC_ 98%; HPLC (C_18_, 35 min): *t_R_* 10.5 min, P_HPLC_ 97%.

#### 4.1.32. 3-[4-[4-[3-(Dimethylamino)propyl]-1-[4-[(4-methylpiperazin-1-yl) Methyl]phenyl] pyrazol-3-yl]phenyl]-*N*,*N*-dimethyl-propan-1-amine (**29**)

General procedure E: 77% yield. The compound was converted to its 4HCl salt (white solid). ^1^H NMR (free amine, CD_3_OD, 300 Mz): *δ* 8.10 (s, 1H), 7.74 (d, *J* = 8.5 Hz, 2H), 7.63 (d, *J* = 8.1 Hz, 2H), 7.42 (d, *J* = 8.5 Hz, 2H), 7.30 (d, *J* = 8.1 Hz, 2H), 3.53 (s, 2H), 2.71–2.65 (m, 4H), 2.49 (br, 8H), 2.42–2.34 (m, 4H), 2.27 (s, 6H), 2.26 (s, 3H), 2.22 (s, 6H), 1.90–1.74 (m, 4H). ^13^C NMR (free amine, CD_3_OD, 75 MHz): *δ* 152.7, 143.0, 140.6, 136.7, 132.6, 131.6, 129.6, 129.1, 128.4, 122.2, 119.6, 63.1, 60.1, 60.0, 55.7, 53.5, 46.0, 45.3 (2C), 34.3, 29.9, 28.7, 23.4. MS (ESI) *m*/*z* 503 [M + H]^+^. P_HPLC_ > 97%. HPLC (C_4_, 35 min): *t_R_* 6.6 min, P_HPLC_ 98%; HPLC (C_18_, 35 min): *t_R_* 11.3 min, P_HPLC_ 97%.

#### 4.1.33. 3-[4-[4-(Dimethylaminomethyl)-1-[3-[(4-methylpiperazin-1-yl)methyl] Phenyl]pyrazol-3-yl]phenyl]-*N*,*N*-dimethyl-propan-1-amine (**30**)

General procedure E: 74% yield. The compound was converted to its 4HCl salt (white solid). ^1^H NMR (free amine, CD_3_OD, 300 Mz): *δ* 8.23 (s, 1H), 7.81 (s, 1H), 7.73–7.69 (m, 3H), 7.43 (t, *J* = 7.8 Hz, 1H), 7.31 (d, *J* = 8.1 Hz, 2H), 7.27 (d, *J* = 8.0 Hz, 1H), 3.58 (s, 2H), 3.52 (s, 2H), 2.68 (t, *J* = 7.6 Hz, 2H), 2.49–2.43 (m, 10H), 2.32 (s, 6H), 2.26 (s, 3H), 2.24 (s, 6H), 1.92–1.82 (m, 2H). ^13^C NMR (free amine, CD_3_OD, 75 MHz): *δ* 153.9, 143.0, 141.3, 140.6, 132.2, 130.5, 130.4, 129.6, 129.6, 128.5, 120.8, 118.9, 118.7, 63.4, 59.9, 55.7, 54.0, 53.5, 45.9, 45.2, 45.1, 34.2, 29.7. MS (ESI) *m*/*z* 475 [M + H]^+^. P_HPLC_ > 97%. HPLC (C_4_, 35 min): *t_R_* 5.1 min, P_HPLC_ 97%; HPLC (C_18_, 35 min): *t_R_* 10.9 min, P_HPLC_ 98%.

#### 4.1.34. 3-[4-[4-[3-(Dimethylamino)propyl]-1-[3-[(4-methylpiperazin-1-yl)methyl] Phenyl] pyrazol-3-yl]phenyl]-*N*,*N*-dimethyl-propan-1-amine (**31**)

General procedure E: 82% yield. The compound was converted to its 4HCl salt (white solid). ^1^H NMR (free amine, CD_3_OD, 300 Mz): *δ* 8.12 (s, 1H), 7.78 (s, 1H), 7.70–7.67 (m, 1H), 7.63 (d, *J* = 8.1 Hz, 2H), 7.41 (t, *J* = 7.8 Hz, 1H), 7.30 (d, *J* = 8.1 Hz, 2H), 7.24 (d, *J* = 7.6 Hz, 1H), 3.56 (s, 2H), 2.72–2.65 (m, 4H), 2.50 (br, 8H), 2.43–2.35 (m, 4H), 2.27 (s, 6H), 2.25 (s, 3H), 2.23 (s, 6H), 1.90–1.77 (m, 4H). ^13^C NMR (free amine, CD_3_OD, 75 MHz): *δ* 152.8, 142.9, 141.4, 140.5, 132.6, 130.5, 129.6, 129.1, 128.4, 128.2, 122.2, 120.6, 118.7, 63.5, 60.1, 60.0, 55.7, 53.6, 46.0, 45.3 (2C), 34.3, 29.9, 28.7, 23.4. MS (ESI) *m*/*z* 503 [M + H]^+^. P_HPLC_ > 97%. HPLC (C_4_, 35 min): *t_R_* 7.9 min, P_HPLC_ 97%; HPLC (C_18_, 35 min): *t_R_* 11.6 min, P_HPLC_ 97%.

#### 4.1.35. *N*′-[[4-[4-(Dimethylaminomethyl)-3-[4-[3-(dimethylamino)propyl]phenyl] Pyrazol-1-yl]phenyl]methyl]-*N*,*N*,*N*′-trimethyl-propane-1,3-diamine (**32**)

General procedure E: 62% yield. The compound was converted to its 4HCl salt (white solid). ^1^H NMR (free amine, CD_3_OD, 300 Mz): *δ* 8.23 (s, 1H), 7.78 (d, *J* = 8.5 Hz, 2H), 7.71 (d, *J* = 8.2 Hz, 2H), 7.46 (d, *J* = 8.5 Hz, 2H), 7.32 (t, *J* = 8.1 Hz, 2H), 3.57 (s, 2H), 3.55 (s, 2H), 2.70 (t, *J* = 7.6 Hz, 2H), 2.52–2.44 (m, 6H), 2.36 (s, 6H), 2.34 (s, 6H), 2.25 (s, 9H), 1.94–1.83 (m, 2H), 1.82–1.72 (m, 2H). ^13^C NMR (free amine, CD_3_OD, 75 MHz): *δ* 154.0, 143.0, 140.4, 137.8, 132.2, 131.6, 130.5, 129.7, 129.6, 119.9, 118.6, 62.4, 59.9, 58.6, 56.0, 53.9, 45.1, 45.1, 45.1, 42.3, 34.2, 29.6, 25.2. MS (ESI) *m*/*z* 491 [M + H]^+^. P_HPLC_ > 95%. HPLC (C_4_, 35 min): *t_R_* 8.1 min, P_HPLC_ 99%; HPLC (C_18_, 35 min): *t_R_* 10.3 min, P_HPLC_ 95%.

#### 4.1.36. *N*′-[[4-[4-[3-(Dimethylamino)propyl]-3-[4-[3-(dimethylamino)propyl]phenyl] Pyrazol-1-yl]phenyl]methyl]-*N*,*N*,*N*′-trimethyl-propane-1,3-diamine (**33**)

General procedure E: 41% yield. The compound was converted to its 4HCl salt (white solid). ^1^H NMR (free amine, CD_3_OD, 300 Mz): *δ* 8.11 (s, 1H), 7.74 (d, *J* = 8.6 Hz, 2H), 7.63 (d, *J* = 8.2 Hz, 2H), 7.43 (d, *J* = 8.6 Hz, 2H), 7.30 (d, *J* = 8.2 Hz, 2H), 3.54 (s, 2H), 2.72–2.65 (m, 4H), 2.44–2.33 (m, 8H), 2.25 (s, 12H), 2.22 (s, 3H), 2.20 (s, 6H), 1.90–1.67 (m, 6H). ^13^C NMR (free amine, CD_3_OD, 75 MHz): *δ* 152.8, 143.0, 140.5, 137.5, 132.6, 131.6, 129.6, 129.1, 128.4, 122.3, 119.6, 62.4, 60.2, 60.1, 58.6, 56.2, 45.4 (3C), 42.3, 34.3, 30.0, 28.8, 25.7, 23.4. MS (ESI) *m*/*z* 519 [M + H]^+^. P_HPLC_ > 97%. HPLC (C_4_, 35 min): *t_R_* 8.3 min, P_HPLC_ 97%; HPLC (C_18_, 35 min): *t_R_* 11.3 min, P_HPLC_ 97%.

#### 4.1.37. *N*′-[[3-[4-(Dimethylaminomethyl)-3-[4-[3-(dimethylamino)propyl]phenyl] Pyrazol-1-yl]phenyl]methyl]-*N*,*N*,*N*′-trimethyl-propane-1,3-diamine (**34**)

General procedure E: 60% yield. The compound was converted to its 4HCl salt (white solid). ^1^H NMR (free amine, CD_3_OD, 300 Mz): *δ* 8.22 (s, 1H), 7.81 (s, 1H), 7.73–7.68 (m, 3H), 7.43 (t, *J* = 7.8 Hz, 1H), 7.29 (d, *J* = 8.3 Hz, 2H), 7.27 (d, *J* = 7.8 Hz, 1H), 3.58 (s, 2H), 3.52 (s, 2H), 2.67 (t, *J* = 7.6 Hz, 2H), 2.45–2.32 (m, 6H), 2.24–2.22 (m, 21H), 1.89–1.69 (m, 4H). ^13^C NMR (free amine, CD_3_OD, 75 MHz): *δ* 153.9, 143.2, 141.6, 141.3, 132.2, 130.5, 130.3, 129.6, 129.5, 128.4, 120.7, 118.8, 118.7, 62.9, 60.1, 58.6, 56.3, 54.0, 45.4 (2C), 45.2, 42.5, 34.4, 30.0, 25.8. MS (ESI) *m*/*z* 491 [M + H]^+^. P_HPLC_ > 96%. HPLC (C_4_, 35 min): *t_R_* 8.0 min, P_HPLC_ 96%; HPLC (C_18_, 35 min): *t_R_* 10.6 min, P_HPLC_ 96%.

#### 4.1.38. *N*′-[[3-[4-[3-(Dimethylamino)propyl]-3-[4-[3-(dimethylamino)propyl]phenyl] Pyrazol-1-yl]phenyl]methyl]-*N*,*N*,*N*′-trimethyl-propane-1,3-diamine (**35**)

General procedure E: 62% yield. The compound was converted to its 4HCl salt (white solid). ^1^H NMR (free amine, CD_3_OD, 300 Mz): *δ* 8.12 (s, 1H), 7.79 (s, 1H), 7.70–7.67 (m, 1H), 7.63 (d, *J* = 8.1 Hz, 2H), 7.42 (t, *J* = 7.8 Hz, 1H), 7.30 (d, *J* = 8.1 Hz, 2H), 7.24 (d, *J* = 7.6 Hz, 1H), 3.57 (s, 2H), 2.72–2.65 (m, 4H), 2.45–2.31 (m, 8H), 2.23 (s, 9H), 2.21 (s, 6H), 2.19 (s, 6H), 1.89–1.67 (m, 6H). ^13^C NMR (free amine, CD_3_OD, 75 MHz): *δ* 152.8, 143.1, 141.5, 141.4, 132.7, 130.4, 129.6, 129.1, 128.4, 128.1, 122.3, 120.6, 118.5, 62.9, 60.2, 60.1, 58.6, 56.3, 45.4 (3C), 42.5, 34.4, 30.1, 28.9, 25.8, 23.5. MS (ESI) *m*/*z* 519 [M + H]^+^. P_HPLC_ > 97%. HPLC (C_4_, 35 min): *t_R_* 7.5 min, P_HPLC_ 98%; HPLC (C_18_, 35 min): *t_R_* 11.4 min, P_HPLC_ 97%.

#### 4.1.39. Methyl 3-[2-[1-(4-Cyanophenyl)ethylidene]hydrazino]benzoate (**38**)

A mixture of 4′-cyanoacetophenone **36** (5 g, 34.4 mmol) and **4** (10 g, 49.3 mmol) in methanol (90 mL) was refluxed for 6 h. The reaction mixture was then stirred at rt for 18 h. The solid was collected by filtration, washed with methanol, and dried to give 7.34 g (73%) of the product as a yellow solid. ^1^H NMR (CD_3_SOCD_3_, 300 Mz): *δ* 9.80 (s, 1H), 7.94 (d, *J* = 8.6 Hz, 2H), 7.87–7.86 (m, 1H), 7.82 (d, *J* = 8.6 Hz, 2H), 7.58–7.54 (m, 1H), 7.42–7.35 (m, 2H), 3.85 (s, 3H), 2.27 (s, 3H). ^13^C NMR (CD_3_SOCD_3_, 75 MHz): *δ* 166.5, 145.8, 143.3, 139.7, 132.2, 130.4, 129.4, 125.8, 120.1, 119.1, 117.4, 113.8, 109.5, 52.1, 12.6. MS (ESI) *m*/*z* 294 [M + H]^+^.

#### 4.1.40. Methyl 3-[2-[1-(3-Cyanophenyl)ethylidene]hydrazino]benzoate (**39**)

A solution of 3’-cyanoacetophenone **37** (3.5 g, 24.1 mmol) and **4** (5.8 g, 28.6 mmol) in methanol (30 mL) was stirred at rt for 1 h, heated at reflux for 6 h and then stirred at rt for 18 h. The solid was collected by filtration, washed with methanol, and dried to give 5.29 g (75%) of the product as a white solid. ^1^H NMR (CD_3_SOCD_3_, 300 Mz): *δ* 9.71 (s, 1H), 8.16–8.10 (m, 2H), 7.84 (dd, *J* = 1.1, 1.1 Hz, 1H), 7.75 (ddd, *J* = 7.7, 1.2, 1.2 Hz, 1H), 7.62–7.56 (m, 2H), 7.39–7.38 (m, 2H), 3.85 (s, 3H), 2.29 (s, 3H). ^13^C NMR (CD_3_SOCD_3_, 75 MHz): *δ* 166.5, 145.9, 140.2, 139.7, 131.0, 130.3, 129.7, 129.6, 129.5, 128.7, 119.9, 118.9, 117.3, 113.7, 111.6, 52.1, 12.8. MS (ESI) *m*/*z* 292 [M − H]^+^.

#### 4.1.41. Methyl 3-[3-(4-Cyanophenyl)-4-formyl-pyrazol-1-yl]benzoate (**40**)

Dimethylformamide (26.9 mL, 348 mmol) was cooled to −5 °C with a salt/ice bath. Phosphorus oxychloride (15.3 g, 9.28 mL, 99.6 mmol) was added dropwise while maintaining the temperature below 0 °C. The mixture was stirred at −5 °C for 40 min and **38** (7.3 g, 24.9 mmol) was slowly added. The reaction mixture was allowed to warm to rt. After 1 h, the mixture was heated at 50 °C for 4 h. The mixture was then poured on water and stirred for 2 h. The solid was collected by filtration, washed with a mixture of methanol and ether (1/3), and dried to give 7.84 g (95%) of the product as a white solid. ^1^H NMR (CF_3_COOD, 300 Mz): *δ* 9.90 (s, 1H), 9.81 (s, 1H), 8.44 (s, 1H), 8.20 (d, *J* = 7.7 Hz, 1H), 7.97–7.94 (m, 3H), 7.86 (d, *J* = 7.9 Hz, 2H), 7.70–7.65 (m, 1H), 4.04 (s, 3H). ^13^C NMR (CF_3_COOD, 75 MHz): *δ* 190.7, 171.2, 156.9, 140.4, 138.7, 136.6, 135.1, 133.5, 133.2, 132.7, 132.2, 128.6, 124.9, 123.8, 55.3. MS (ESI) *m*/*z* 332 [M + H]^+^.

#### 4.1.42. Methyl 3-[3-(3-Cyanophenyl)-4-formyl-pyrazol-1-yl]benzoate (**41**)

General procedure A: 95% yield (white solid). ^1^H NMR (CDCl_3_, 300 Mz): *δ* 10.07 (s, 1H), 8.64 (s, 1H), 8.43 (dd, *J* = 1.8, 1.8 Hz, 1H), 8.29 (dd, *J* = 1.4, 1.4 Hz, 1H), 8.22 (ddd, *J* = 7.9, 1.3, 1.3 Hz, 1H), 8.09 (ddd, *J* = 7.9, 1.3, 1.3 Hz, 1H), 8.07–8.03 (m, 1H), 7.75 (ddd, *J* = 7.8, 1.6, 1.6 Hz, 1H), 7.66–7.59 (m, 2H), 3.99 (s, 3H). ^13^C NMR (CDCl_3_, 75 MHz): *δ* 183.6, 165.9, 151.8, 139.0, 133.6, 133.3, 132.8, 132.7, 132.5, 132.1, 130.2, 129.6, 129.3, 124.0, 123.1, 120.4, 118.6, 113.1, 52.8. MS (ESI) *m*/*z* 332 [M + H]^+^.

#### 4.1.43. Methyl 3-[3-(4-Cyanophenyl)-4-methyl-pyrazol-1-yl]benzoate (**42**)

A mixture of **40** (7.76 g, 23.4 mmol), triethylsilane (9.45 mL, 58.5 mmol), and trifluoroacetic acid (26.1 mL, 351 mmol) was stirred vigorously at rt for 24 h. The reaction mixture was then evaporated to dryness. The residue was purified by column chromatography (DCM/MeOH = 95:5) to give 6.44 g (87%) of the product as a white solid. ^1^H NMR (CDCl_3_, 300 Mz): *δ* 8.34–8.33 (m, 1H), 8.01–7.93 (m, 4H), 7.89–7.88 (m, 1H), 7.75–7.72 (m, 2H), 7.54 (dd, *J* = 8.0, 8.0 Hz, 1H), 3.96 (s, 3H), 2.35 (s, 3H). ^13^C NMR (CDCl_3_, 75 MHz): *δ* 166.4, 149.9, 140.1, 138.3, 132.5, 131.7, 129.8, 127.9, 127.8, 127.5, 123.1, 119.4, 119.1, 117.4, 111.2, 52.6, 10.6. MS (ESI) *m*/*z* 318 [M + H]^+^.

#### 4.1.44. Methyl 3-[3-(3-Cyanophenyl)-4-methyl-pyrazol-1-yl]benzoate (**43**)

A mixture of **41** (2.5 g, 7.55 mmol), triethylsilane (2.1 g, 2.92 mL, 18.1 mmol), and trifluoroacetic acid (12.8 g, 8.33 mL, 112 mmol) was vigorously stirred at rt for 24 h. It was then evaporated to dryness. The residue was purified by column chromatography (DCM) to give 2.07 g (86%) of the product as a white solid. ^1^H NMR (CDCl_3_, 300 Mz): *δ* 8.33 (s, 1H), 8.11 (s, 1H), 8.05 (d, *J* = 7.8 Hz, 1H), 8.00–7.94 (m, 2H), 7.88 (s, 1H), 7.64 (d, *J* = 7.7 Hz, 1H), 7.58–7.52 (m, 2H), 3.97 (s, 3H), 2.34 (s, 3H). ^13^C NMR (CDCl_3_, 75 MHz): *δ* 166.3, 149.5, 139.9, 134.9, 131.5, 131.5, 131.0, 130.8, 129.7, 129.4, 127.6, 127.3, 122.9, 119.2, 118.9, 116.9, 112.7, 52.4, 10.4. MS (ESI) *m*/*z* 318 [M + H]^+^.

#### 4.1.45. Methyl 3-[3-[4-(Aminomethyl)phenyl]-4-methyl-pyrazol-1-yl]benzoate (**44**)

To a solution of **42** (6.37 g, 20.1 mmol) in anhydrous THF (90 mL) under nitrogen was added BH_3_-THF (1 M in THF, 30.1 mL, 30.1 mmol). The reaction mixture was refluxed for 3 h. Methanol (25 mL) was then slowly added. HCl (4 M in dioxane, 27.6 mL, 110 mmol) was then added and the mixture was refluxed for 90 min. The solvent was evaporated. Ethyl acetate and water were added to the residue and the pH of the mixture was brought to 10 by the addition of potassium carbonate. The layers were separated. The aqueous layer was extracted twice with ethyl acetate. The combined organic layers were washed with brine, dried, and evaporated. The residue was purified by column chromatography (DCM/MeOH-NH_3_ sat = 95:5) to give 3.3 g (51%) of the product as a colorless oil. ^1^H NMR (CDCl_3_, 300 Mz): *δ* 8.26–8.25 (m, 1H), 7.89–7.85 (m, 1H), 7.82–7.79 (m, 1H), 7.71–7.68 (m, 3H), 7.38 (dd, *J* = 7.9, 7.9 Hz, 1H), 7.31 (d, *J* = 8.2 Hz, 2H), 3.85 (s, 3H), 3.82 (s, 2H), 2.21 (s, 3H), 1.53 (s, 2H). ^13^C NMR (CDCl_3_, 75 MHz): *δ* 166.1, 151.5, 142.7, 139.9, 131.8, 131.1, 129.2, 127.4, 126.9, 126.8, 126.4, 122.4, 118.6, 116.4, 52.1, 46.0, 10.1. MS (ESI) *m*/*z* 322 [M + H]^+^.

#### 4.1.46. Methyl 3-[3-[3-(Aminomethyl)phenyl]-4-methyl-pyrazol-1-yl]benzoate (**45**)

To a solution of **43** (3.03 g, 9.53 mmol) in anhydrous THF (60 mL) under nitrogen was added BH_3_-THF (1 M in THF, 13.3 mL, 13.3 mmol). The reaction mixture was refluxed for 2 h 30. Methanol (25 mL) was then slowly added, followed by HCl (4 M in dioxane, 13.1 mL, 52.4 mmol), and the mixture was refluxed for 90 min. The solvent was evaporated. Ethyl acetate and water were added to the residue and the pH of the mixture was brought to 10 by the addition of potassium carbonate. The layers were separated. The aqueous layer was extracted twice with ethyl acetate. The combined organic layers were washed with brine, dried, and evaporated. The residue was purified by column chromatography (DCM/MeOH-NH_3_ sat = 95:5) to give 2.31 g (75%) of the product as a colorless oil. ^1^H NMR (CDCl_3_, 300 Mz): *δ* 8.32 (dd, *J* = 1.8, 1.8 Hz, 1H), 7.98–7.95 (m, 1H), 7.89 (ddd, *J* = 7.8, 1.3, 1.3 Hz, 1H), 7.81 (s, 1H), 7.75 (s, 1H), 7.62 (d, *J* = 7.7 Hz, 1H), 7.48 (dd, *J* = 7.9, 7.9 Hz, 1H), 7.40 (dd, *J* = 7.6, 7.6 Hz, 1H), 7.30 (d, *J* = 7.6 Hz, 1H), 3.92 (s, 5H), 2.29 (s, 3H), 1.69 (s, 2H). ^13^C NMR (CDCl_3_, 75 MHz): *δ* 166.4, 152.0, 143.6, 140.2, 133.8, 131.4, 129.5, 128.7, 127.1, 126.8, 126.6, 126.3, 126.1, 122.9, 119.1, 116.8, 52.3, 46.6, 10.3. MS (ESI) *m*/*z* 322 [M + H]^+^.

#### 4.1.47. Methyl 3-[3-[4-(Dimethylaminomethyl)phenyl]-4-methyl-pyrazol-1-yl]benzoate (**46**)

General procedure B: 46% yield (colorless oil). ^1^H NMR (CD_3_OD, 300 Mz): *δ* 8.24–8.22 (m, 1H), 7.83–7.83 (m, 1H), 7.82–7.78 (m, 1H), 7.74–7.71 (m, 1H), 7.64 (d, *J* = 8.2 Hz, 2H), 7.37 (dd, *J* = 7.9, 7.9 Hz, 1H), 7.28 (d, *J* = 8.2 Hz, 2H), 3.82 (s, 3H), 3.38 (s, 2H), 2.27 (s, 6H), 2.14 (s, 3H). ^13^C NMR (CD_3_OD, 75 MHz): *δ* 167.4, 152.5, 141.2, 138.2, 133.9, 132.4, 130.6, 128.7, 128.4, 127.5, 123.3, 119.8, 117.8, 65.5, 52.8, 45.2, 10.5. MS (ESI) *m*/*z* 350 [M + H]^+^.

#### 4.1.48. Methyl 3-[3-[3-(Dimethylaminomethyl)phenyl]-4-methyl-pyrazol-1-yl] Benzoate (**47**)

General procedure B: 93% yield (colorless oil). ^1^H NMR (CDCl_3_, 300 Mz): *δ* 8.31 (dd, *J* = 1.8, 1.8 Hz, 1H), 7.96–7.92 (m, 1H), 7.88–7.85 (m, 1H), 7.78–7.77 (m, 1H), 7.73–7.72 (m, 1H), 7.66–7.64 (m, 1H), 7.44 (dd, *J* = 7.9, 7.9 Hz, 1H), 7.38 (dd, *J* = 7.6, 7.6 Hz, 1H), 7.33–7.30 (m, 1H), 3.90 (s, 3H), 3.48 (s, 2H), 2.27–2.25 (m, 9H). ^13^C NMR (CDCl_3_, 75 MHz): *δ* 166.3, 151.9, 140.1, 139.1, 133.4, 131.3, 129.4, 128.5, 128.4, 128.3, 126.9, 126.6, 126.3, 122.7, 118.9, 116.7, 64.3, 52.2, 45.3, 10.3. MS (ESI) *m*/*z* 350 [M + H]^+^.

#### 4.1.49. [3-[3-[4-(Dimethylaminomethyl)phenyl]-4-methyl-pyrazol-1-yl]phenyl] Methanol (**48**)

General procedure C: 85% yield (colorless oil). ^1^H NMR (CD_3_OD, 300 Mz): *δ* 7.98–7.98 (m, 1H), 7.76–7.75 (m, 1H), 7.72–7.69 (m, 2H), 7.63–7.60 (m, 1H), 7.42–7.35 (m, 3H), 7.26–7.24 (m, 1H), 4.66 (s, 2H), 3.49 (s, 2H), 2.24 (s, 9H). ^13^C NMR (CD_3_OD, 75 MHz): *δ* 152.5, 144.6, 141.3, 138.0, 134.2, 130.8, 130.5, 129.2, 128.6, 125.5, 118.4, 118.0, 117.5, 64.7, 64.5, 45.1, 10.3. MS (ESI) *m*/*z* 322 [M + H]^+^.

#### 4.1.50. [3-[3-[3-(Dimethylaminomethyl)phenyl]-4-methyl-pyrazol-1-yl]phenyl] Methanol (**49**)

General procedure C: 88% yield (colorless oil). ^1^H NMR (CDCl_3_, 300 Mz): *δ* 7.72 (s, 1H), 7.67–7.64 (m, 3H), 7.58–7.55 (m, 1H), 7.39 (dd, *J* = 7.6, 7.6 Hz, 1H), 7.33–7.28 (m, 2H), 7.13 (d, *J* = 7.6 Hz, 1H), 4.62 (s, 2H), 4.48 (s, 1H), 3.48 (s, 2H), 2.26 (s, 3H), 2.23 (s, 6H). ^13^C NMR (CDCl_3_, 75 MHz): *δ* 151.4, 143.2, 140.1, 138.4, 133.7, 129.3, 128.6, 128.5, 128.5, 127.2, 126.5, 124.1, 117.3, 116.7, 116.2, 64.2, 45.2 (2C), 10.3. MS (ESI) *m*/*z* 322 ([M + H]^+^.

#### 4.1.51. 3-[3-[4-(Dimethylaminomethyl)phenyl]-4-methyl-pyrazol-1-yl] Benzaldehyde (**50**)

General procedure D: 56% yield (colorless oil). ^1^H NMR (CDCl_3_, 300 Mz): *δ* 9.99 (s, 1H), 8.15–8.14 (m, 1H), 7.99–7.95 (m, 1H), 7.78–7.77 (m, 1H), 7.72 (d, *J* = 8.2 Hz, 2H), 7.68–7.65 (m, 1H), 7.52 (dd, *J* = 7.8, 7.8 Hz, 1H), 7.36 (d, *J* = 8.2 Hz, 2H), 3.44 (s, 2H), 2.26 (s, 3H), 2.23 (s, 6H). ^13^C NMR (CDCl_3_, 75 MHz): *δ* 191.5, 152.0, 140.6, 138.4, 137.3, 132.2, 130.1, 129.3, 127.4, 126.9 (2C), 123.8, 118.3, 117.0, 64.0, 45.3, 10.3. MS (ESI) *m*/*z* 320 [M + H]^+^.

#### 4.1.52. 3-[3-[3-(Dimethylaminomethyl)phenyl]-4-methyl-pyrazol-1-yl]benzaldehyde (**51**)

General procedure D: 61% yield (colorless oil). ^1^H NMR (CDCl_3_, 300 Mz): *δ* 10.03 (s, 1H), 8.18 (dd, *J* = 1.8, 1.8 Hz, 1H), 8.03–7.99 (m, 1H), 7.82–7.81 (m, 1H), 7.73 (dd, *J* = 1.5, 1.5 Hz, 1H), 7.70 (ddd, *J* = 7.6, 1.2, 1.2 Hz, 1H), 7.65 (ddd, *J* = 7.5, 1.5, 1.5 Hz, 1H), 7.55 (dd, *J* = 7.8, 7.8 Hz, 1H), 7.40 (dd, *J* = 7.6, 7.6 Hz, 1H), 7.33 (ddd, *J* = 7.7, 1.5, 1.5 Hz, 1H), 3.49 (s, 2H), 2.29–2.27 (m, 9H). ^13^C NMR (CDCl_3_, 75 MHz): *δ* 191.6, 152.3, 140.8, 139.2, 137.5, 133.3, 130.1, 128.7, 128.5, 128.3, 127.0, 126.9, 126.4, 123.9, 118.5, 117.1, 64.4, 45.4, 10.3. MS (ESI) *m*/*z* 320 [M + H]^+^.

#### 4.1.53. 3-(Dimethylamino)propanoic Acid Hydrochloride (**52**)

A mixture of beta-alanine (5.4 g, 60.6 mmol), formic acid (40 mL, 1060 mmol), and 37% formaldehyde in water (13 mL, 173 mmol) was refluxed for 15 h. A total of 37% HCl (12 mL) was added, and the reaction mixture was evaporated. The residue was washed with a mixture of ethyl acetate and methanol (4:1), collected by filtration, and dried to give 8.07 g (87%) of the product as a white solid. ^1^H NMR (CD_3_OD, 300 Mz): *δ* 3.44 (t, *J* = 6.9 Hz, 2H), 2.93 (s, 6H), 2.88 (t, *J* = 6.9 Hz, 2H). ^13^C NMR (CD_3_OD, 75 MHz): *δ* 173.3, 54.6, 43.7, 29.8. MS (ESI) *m*/*z* 117 [M]^+^.

#### 4.1.54. Methyl 3-[3-(4-{[3-(Dimethylamino)propanamido]methyl}phenyl)-4-methyl- 1H-pyrazol-1-yl] Benzoate (**53**)

To a mixture of **44** (1.87 g, 5.83 mmol), **52** (0.98 g, 6.41 mmol), hydroxybenzotriazole hydrate (1.02 g, 7.58 mmol) and triethylamine (3.25 mL, 23.3 mmol) in methylene chloride (60 mL) under nitrogen at rt was added 1-ethyl-3-(3-dimethylaminopropyl)carbodiimide hydrochloride (1.34 g, 7 mmol). The mixture was stirred for 20 h. A total of 10% K_2_CO_3_ was added, and the layers were separated. The aqueous layer was extracted twice with methylene chloride. The combined organic layers were washed with brine, dried, and evaporated. The residue was purified by column chromatography (DCM/MeOH/NH_4_OH = 475:25:1) to give 1.77 g (72%) of the product as a white solid. ^1^H NMR (CDCl_3_, 300 Mz): *δ* 8.75 (br, 1H), 8.33–8.32 (m, 1H), 8.01–7.97 (m, 1H), 7.93–7.89 (m, 1H), 7.85–7.84 (m, 1H), 7.75 (d, *J* = 8.2 Hz, 2H), 7.51 (dd, *J* = 7.9, 7.9 Hz, 1H), 7.34 (d, *J* = 8.2 Hz, 2H), 4.49 (d, *J* = 5.7 Hz, 2H), 3.95 (s, 3H), 2.59 (t, *J* = 5.9 Hz, 2H), 2.44 (t, *J* = 6.2 Hz, 2H), 2.31 (s, 3H), 2.25 (s, 6H). ^13^C NMR (CDCl_3_, 75 MHz): *δ* 172.6, 166.6, 151.8, 140.3, 138.5, 132.5, 131.5, 129.7, 127.9, 127.6, 127.2, 126.9, 123.0, 119.2, 116.9, 55.4, 52.5, 44.7, 42.9, 33.0, 10.4. MS (ESI) *m*/*z* 421 [M + H]^+^.

#### 4.1.55. Methyl 3-[3-[3-[[3-(Dimethylamino)propanoylamino]methyl]phenyl]-4-methyl-pyrazol-1-yl]benzoate (**54**)

To a mixture of **45** (0.5 g, 1.56 mmol), **52** (0.26 g, 1.71 mmol), hydroxybenzotriazole hydrate (0.27 g, 2.02 mmol) and triethylamine (0.86 mL, 6.22 mmol) in methylene chloride (15 mL) under nitrogen at rt was added 1-ethyl-3-(3-dimethylaminopropyl)carbodiimide hydrochloride (0.36 g, 1.87 mmol). The mixture was stirred for 39 h. A total of 10% K_2_CO_3_ was added, and the layers were separated. The aqueous layer was extracted with methylene chloride. The combined organic layers were washed with brine, dried, and evaporated. The residue was purified by column chromatography (DCM/MeOH/NH_4_OH = 95:5:0.3) to give 535 mg (82%) of the product as a colorless oil. ^1^H NMR (CDCl_3_, 300 Mz): *δ* 8.69–8.68 (m, 1H), 8.27 (dd, *J* = 1.9, 1.9 Hz, 1H), 7.91–7.87 (m, 1H), 7.84 (ddd, *J* = 7.8, 1.3, 1.3 Hz, 1H), 7.78–7.77 (m, 1H), 7.67 (s, 1H), 7.60 (d, *J* = 7.7 Hz, 1H), 7.43 (dd, *J* = 7.9, 7.9 Hz, 1H), 7.34 (dd, *J* = 7.6, 7.6 Hz, 1H), 7.22 (d, *J* = 7.7 Hz, 1H), 4.47 (d, *J* = 5.7 Hz, 2H), 3.88 (s, 3H), 2.56 (t, *J* = 6.0 Hz, 2H), 2.39 (t, *J* = 6.3 Hz, 2H), 2.24 (s, 3H), 2.19 (s, 6H). ^13^C NMR (CDCl_3_, 75 MHz): *δ* 172.3, 166.3, 151.7, 140.1, 139.1, 133.7, 131.3, 129.4, 128.6, 127.0, 126.7 (2C), 126.4, 126.2, 122.6, 118.9, 116.7, 55.1, 52.2, 44.5, 42.9, 32.9, 10.2. MS (ESI) *m*/*z* 421 [M + H]^+^.

#### 4.1.56. (3-{3-[4-({[3-(Dimethylamino)propyl]amino}methyl)phenyl]-4-methy-1H-pyrazol-1-yl}phenyl)methanol (**55**)

To a suspension of aluminum chloride (2.39 g, 18 mmol) in anhydrous THF (50 mL) at 0 °C under nitrogen was added dropwise lithium aluminum hydride (1 M in THF, 18 mL, 18 mmol). The mixture was stirred for 20 min and a solution of **53** (1.68 g, 3.99 mmol) in anhydrous THF (50 mL) was added dropwise. The reaction mixture was stirred at 0 °C for 30 min and then allowed to warm to rt. After 20 h, the solution was poured on ice. Ethyl acetate and K_2_CO_3_ were added, and the mixture was stirred for 15 min. The solid was filtered off. The layers were separated, and the organic layer was washed with brine, dried, and evaporated. The residue was purified by column chromatography (DCM/MeOH/NH_4_OH = 9:1:0.1) to give 740 mg (49%) of the product as a colorless oil. ^1^H NMR (CDCl_3_, 300 Mz): *δ* 7.75–7.70 (m, 4H), 7.61–7.58 (m, 1H), 7.40–7.34 (m, 3H), 7.22–7.19 (m, 1H), 4.69 (s, 2H), 3.79 (s, 2H), 3.00 (br, 2H), 2.66 (t, *J* = 7.0 Hz, 2H), 2.30 (t, *J* = 7.0 Hz, 2H), 2.27 (s, 3H), 2.20 (s, 6H), 1.68 (quint, *J* = 7.2 Hz, 2H). ^13^C NMR (CDCl_3_, 75 MHz): *δ* 151.5, 143.2, 140.3, 139.3, 132.6, 129.5, 128.4, 127.7, 127.2, 124.2, 117.5, 117.0, 116.3, 64.6, 58.2, 53.7, 47.9, 45.5, 27.6, 10.4. MS (ESI) *m*/*z* 379 [M + H]^+^.

#### 4.1.57. [3-[3-[3-[[3-(Dimethylamino)propylamino]methyl]phenyl]-4-methyl-pyrazol-1-yl]phenyl]methanol (**56**)

To a suspension of aluminum chloride (1.53 g, 11.5 mmol) in anhydrous THF (40 mL) at 0 °C under nitrogen was added dropwise LAH (1 M in THF, 11.5 mL, 11.5 mmol). The mixture was stirred for 20 min and a solution of **54** (1.07 g, 2.54 mmol) in anhydrous THF (40 mL) was added dropwise. The reaction mixture was stirred at 0 °C for 30 min and then allowed to warm to rt. After 20 h, the solution was slowly poured on ice. Ethyl acetate and K_2_CO_3_ were added, and the mixture was stirred for 15 min. The solid was filtered off. The layers were separated, and the organic layer was washed with brine, dried, and evaporated. The residue was purified by column chromatography (DCM/MeOH/NH_4_OH = 9:1:0.1) to give 737 mg (77%) of the product as a colorless oil. ^1^H NMR (CDCl_3_, 300 Mz): *δ* 7.71–7.70 (m, 3H), 7.63–7.60 (m, 1H), 7.57–7.54 (m, 1H), 7.39–7.26 (m, 3H), 7.17 (d, *J* = 7.7 Hz, 1H), 4.64 (s, 2H), 3.80 (s, 2H), 3.41 (br, 2H), 2.65 (t, *J* = 7.0 Hz, 2H), 2.29 (t, *J* = 7.1 Hz, 2H), 2.25 (s, 3H), 2.16 (s, 6H), 1.71–1.61 (m, 2H). ^13^C NMR (CDCl_3_, 75 MHz): *δ* 151.4, 143.5, 140.2, 133.9, 129.3, 128.6, 127.5, 127.4, 127.2, 126.3, 124.1, 117.3, 116.8, 116.2, 64.2, 58.2, 53.9, 47.8, 45.4, 27.6, 10.3. MS (ESI) *m*/*z* 379 [M + H]^+^.

#### 4.1.58. (3-{3-[4-({[3-(Dimethylamino)propyl](methyl)amino}methyl)phenyl]-4-methyl-1H-pyrazol-1-yl}phenyl)methanol (**57**)

To a mixture of **55** (0.72 g, 1.9 mmol), 37% formaldehyde in water (0.85 mL, 11.4 mmol) and acetic acid (0.65 mL, 11.4 mmol) in methanol (15 mL) were slowly added over 40 min STAB (2.42 g, 11.4 mmol). The mixture was stirred for 1 h and the solvent was evaporated. Ethyl acetate and 10% K_2_CO_3_ were added. After 10 min of stirring, the layers were separated. The aqueous layer was extracted twice with ethyl acetate. The combined organic layers were washed with brine, dried, and evaporated. The residue was purified by column chromatography (DCM/MeOH-NH_3_ sat = 95:5 to 92:8) to give 0.55 g (74%) of the product as a colorless oil. ^1^H NMR (CDCl_3_, 300 Mz): *δ* 7.77–7.70 (m, 4H), 7.63–7.60 (m, 1H), 7.42–7.35 (m, 3H), 7.23–7.21 (m, 1H), 4.72 (s, 2H), 3.51 (s, 2H), 2.92 (br, 1H), 2.40 (t, *J* = 7.3 Hz, 2H), 2.33–2.28 (m, 5H), 2.22 (s, 6H), 2.20 (s, 3H), 1.76–1.66 (m, 2H). ^13^C NMR (CDCl_3_, 75 MHz): *δ* 151.7, 143.0, 140.4, 138.6, 132.5, 129.6, 129.3, 127.5, 127.2, 124.2, 117.6, 117.0, 116.3, 64.8, 62.2, 57.9, 55.6, 45.5, 42.3, 25.6, 10.4. MS (ESI) *m*/*z* 393 [M + H]^+^.

#### 4.1.59. [3-[3-[3-[[3-(Dimethylamino)propyl-methyl-amino]methyl]phenyl]-4-methyl-pyrazol-1-yl]phenyl]methanol (**58**)

To a solution of **56** (0.79 g, 2.08 mmol), 37% formaldehyde in water (0.47 mL, 6.23 mmol) and acetic acid (0.36 mL, 6.23 mmol) in methanol (15 mL) were slowly added over 15 min STAB (1.1 g, 5.19 mmol). The mixture was stirred at rt for 15 h and the solvent was evaporated. Methylene chloride and water were added to the residue. The mixture was brought to pH = 10 with ammonium hydroxide and the layers were separated. The aqueous layer was extracted twice with methylene chloride. The combined organic layers were washed with brine, dried, and evaporated. The residue was purified by column chromatography (DCM/MeOH/NH_4_OH = 95:5:0.5) to give 591 mg (73%) of the product as a colorless oil. ^1^H NMR (CDCl_3_, 300 Mz): *δ* 7.73–7.70 (m, 3H), 7.65–7.62 (m, 1H), 7.57–7.54 (m, 1H), 7.39–7.27 (m, 3H), 7.16 (d, *J* = 7.6 Hz, 1H), 4.95 (br, 1H), 4.64 (s, 2H), 3.52 (s, 2H), 2.39 (t, *J* = 7.2 Hz, 2H), 2.33–2.28 (m, 2H), 2.26 (s, 3H), 2.19 (s, 3H), 2.18 (s, 6H), 1.74–1.64 (m, 2H). ^13^C NMR (CDCl_3_, 75 MHz): *δ* 151.5, 143.5, 140.1, 139.1, 133.6, 129.2, 128.3, 128.2, 127.1, 126.2, 124.0, 117.1, 116.7, 116.1, 64.1, 62.3, 57.7, 55.4, 45.2, 42.2, 25.2, 10.3. MS (ESI) *m*/*z* 393 [M + H]^+^

#### 4.1.60. 3-{3-[4-({[3-(Dimethylamino)propyl](methyl)amino}methyl)phenyl]-4-methyl-1H-pyrazol-1-yl}benzaldehyde (**59**)

General procedure D: 66% yield (colorless oil). ^1^H NMR (CDCl_3_, 300 Mz): *δ* 10.03 (s, 1H), 8.18–8.17 (m, 1H), 8.03–7.99 (m, 1H), 7.82–7.82 (m, 1H), 7.73–7.68 (m, 3H), 7.56 (dd, *J* = 7.8, 7.8 Hz, 1H), 7.38 (d, *J* = 8.2 Hz, 2H), 3.51 (s, 2H), 2.41 (t, *J* = 7.3 Hz, 2H), 2.34–2.29 (m, 5H), 2.22 (s, 6H), 2.20 (s, 3H), 1.75–1.65 (m, 2H). ^13^C NMR (CDCl_3_, 75 MHz): *δ* 191.6, 152.2, 140.8, 139.0, 137.5, 132.0, 130.2, 129.2, 127.4, 127.0, 126.9, 123.9, 118.4, 117.1, 62.2, 57.9, 55.5, 45.5, 42.3, 25.7, 10.4. MS (ESI) *m*/*z* 391 [M + H]^+^.

#### 4.1.61. 3-[3-[3-[[3-(Dimethylamino)propyl-methyl-amino]methyl]phenyl]-4-methyl-pyrazol-1-yl]benzaldehyde (**60**)

General procedure D: 55% yield (colorless oil). ^1^H NMR (CDCl_3_, 300 Mz): *δ* 10.00 (s, 1H), 8.15 (dd, *J* = 1.7, 1.7 Hz, 1H), 8.00–7.96 (m, 1H), 7.79–7.78 (m, 1H), 7.71–7.70 (m, 1H), 7.67 (ddd, *J* = 7.6, 1.2, 1.2 Hz, 1H), 7.62 (ddd, *J* = 7.5, 1.5, 1.5 Hz, 1H), 7.52 (dd, *J* = 7.9, 7.9 Hz, 1H), 7.36 (dd, *J* = 7.6, 7.6 Hz, 1H), 7.31–7.29 (m, 1H), 3.52 (s, 2H), 2.41 (t, *J* = 7.2 Hz, 2H), 2.32–2.27 (m, 2H), 2.27 (s, 3H), 2.19 (s, 9H), 1.73–1.63 (m, 2H). ^13^C NMR (CDCl_3_, 75 MHz): *δ* 191.5, 152.3, 140.7, 139.6, 137.4, 133.2, 130.1, 128.5, 128.4, 128.1, 126.9, 126.2, 123.8, 118.3, 117.0, 62.3, 57.8, 55.6, 45.4, 42.2, 25.6, 10.3. MS (ESI) *m*/*z* 391 [M + H]^+^.

#### 4.1.62. {[4-(1-{3-[(Dimethylamino)methyl]phenyl}-4-methyl-1H-pyrazol-3-yl)phenyl]methyl}dimethylamine (**61**)

General procedure E: 79% yield. The compound was converted to its 2HCl salt (white solid). ^1^H NMR (free amine, CD_3_OD, 300 Mz): *δ* 8.04–8.03 (m, 1H), 7.74–7.72 (m, 3H), 7.69–7.65 (m, 1H), 7.43–7.37 (m, 3H), 7.22–7.20 (m, 1H), 3.49 (s, 2H), 3.48 (s, 2H), 2.28–2.24 (m, 15H). ^13^C NMR (free amine, CD_3_OD, 75 MHz): *δ* 152.6, 141.4, 140.7, 138.3, 134.2, 130.8, 130.5, 129.2, 128.6, 128.2, 120.6, 118.7, 117.6, 64.7, 64.6, 45.3, 45.2, 10.3. MS (ESI) *m*/*z* 349 [M + H]^+^. P_HPLC_ > 96%. HPLC (C_4_, 35 min): *t_R_* 13.4 min, P_HPLC_ 99%; HPLC (C_18_, 35 min): *t_R_* 17.6 min, P_HPLC_ 96%.

#### 4.1.63. Dimethyl({[4-(4-methyl-1-{3-[(4-methylpiperazin-1-yl)methyl]phenyl}-1H-pyrazol-3-yl)phenyl]methyl})amine (**62**)

General procedure E: 61% yield. The compound was converted to its 3HCl salt (white solid). ^1^H NMR (free amine, CD_3_OD, 300 Mz): *δ* 8.04–8.04 (m, 1H), 7.76–7.71 (m, 3H), 7.67–7.64 (m, 1H), 7.42–7.38 (m, 3H), 7.24–7.22 (m, 1H), 3.56 (s, 2H), 3.50 (s, 2H), 2.49 (m, 8H), 2.28–2.25 (m, 12H). ^13^C NMR (free amine, CD_3_OD, 75 MHz): *δ* 152.6, 141.4, 140.5, 138.3, 134.2, 130.8, 130.4, 129.2, 128.7, 128.2, 120.6, 118.7, 117.6, 64.6, 63.5, 55.7, 53.6, 46.0, 45.2, 10.3. MS (ESI) *m*/*z* 404 [M + H]^+^. P_HPLC_ > 96%. HPLC (C_4_, 30 min): *t_R_* 12.1 min, P_HPLC_ 96%; HPLC (C_18_, 30 min): *t_R_* 17.3 min, P_HPLC_ 97%.

#### 4.1.64. [(4-{1-[3-({[3-(Dimethylamino)propyl](methyl)amino}methyl)phenyl]-4-methyl-1H-pyrazol-3-yl}phenyl)methyl]dimethylamine (**63**)

General procedure E: 73% yield. The compound was converted to its 3HCl salt (white solid). ^1^H NMR (free amine, CD_3_OD, 300 Mz): *δ* 8.05–8.05 (m, 1H), 7.77–7.72 (m, 3H), 7.67–7.64 (m, 1H), 7.43–7.38 (m, 3H), 7.24–7.21 (m, 1H), 3.56 (s, 2H), 3.50 (s, 2H), 2.44–2.39 (m, 2H), 2.36–2.31 (m, 2H), 2.29 (s, 3H), 2.25 (s, 6H), 2.23 (s, 3H), 2.22 (s, 6H), 1.76–1.66 (m, 2H). ^13^C NMR (free amine, CD_3_OD, 75 MHz): *δ* 152.6, 141.4, 141.4, 138.3, 134.2, 130.8, 130.4, 129.2, 128.7, 128.1, 120.5, 118.5, 117.5, 64.7, 62.9, 58.6, 56.2, 45.4, 45.2, 42.5, 25.7, 10.4. MS (ESI) *m*/*z* 420 [M + H]^+^. P_HPLC_ > 95%. HPLC (C_4_, 35 min): *t_R_* 12.7 min, P_HPLC_ 98%; HPLC (C_18_, 35 min): *t_R_* 16.6 min, P_HPLC_ 95%.

#### 4.1.65. 1-[3-[1-[3-(Dimethylaminomethyl)phenyl]-4-methyl-pyrazol-3-yl]phenyl]-*N*,*N*-dimethylmethanamine (**64**)

General procedure E: 80% yield. The compound was converted to its 2HCl salt (white solid). ^1^H NMR (free amine, CD_3_OD, 300 Mz): *δ* 8.05 (s, 1H), 7.76–7.72 (m, 2H), 7.70–7.66 (m, 2H), 7.45–7.39 (m, 2H), 7.33–7.30 (m, 1H), 7.23–7.21 (m, 1H), 3.53 (s, 2H), 3.51 (s, 2H), 2.29–2.25 (m, 15H). ^13^C NMR (free amine, CD_3_OD, 75 MHz): *δ* 152.8, 141.4, 140.7, 139.0, 135.0, 130.5, 130.1, 129.9, 129.6, 129.2, 128.2, 127.9, 120.7, 118.8, 117.6, 64.9, 64.7, 45.3, 45.2, 10.3. MS (ESI) *m*/*z* 349 [M + H]^+^. P_HPLC_ > 97%. HPLC (C_4_, 35 min): *t_R_* 9.1 min, P_HPLC_ 98%; HPLC (C_18_, 35 min): *t_R_* 13.7 min, P_HPLC_ 97%.

#### 4.1.66. *N*,*N*-dimethyl-1-[3-[4-methyl-1-[3-[(4-methylpiperazin-1-yl)methyl]phenyl]pyrazol-3-yl]phenyl]methanamine (**65**)

General procedure E: 27% yield. The compound was converted to its 3HCl salt (white solid). ^1^H NMR (free amine, CD_3_OD, 300 Mz): *δ* 8.08–8.07 (m, 1H), 7.79–7.73 (m, 2H), 7.70–7.66 (m, 2H), 7.47–7.40 (m, 2H), 7.36–7.32 (m, 1H), 7.27–7.25 (m, 1H), 3.60 (s, 2H), 3.58 (s, 2H), 2.52 (br, 8H), 2.31–2.28 (m, 12H). ^13^C NMR (free amine, CD_3_OD, 75 MHz): *δ* 152.8, 141.5, 140.5, 138.8, 135.1, 130.5, 130.1, 130.0, 129.6, 129.3, 128.3, 128.1, 120.7, 118.7, 117.6, 64.8, 63.5, 55.7, 53.5, 45.9, 45.2, 10.2. MS (ESI) *m*/*z* 404 [M + H]^+^. P_HPLC_ > 98%. HPLC (C_4_, 35 min): *t_R_* 12.1 min, P_HPLC_ 98%; HPLC (C_18_, 35 min): *t_R_* 18.3 min, P_HPLC_ 98%.

#### 4.1.67. *N*′-[[3-[3-[3-(Dimethylaminomethyl)phenyl]-4-methyl-pyrazol-1-yl]phenyl] methyl]-N,N,N’-trimethyl-propane-1,3-diamine (**66**)

General procedure E: 78% yield. The compound was converted to its 3HCl salt (white solid). ^1^H NMR (salt, CD_3_OD, 300 Mz): *δ* 8.28–8.26 (m, 2H), 8.09 (s, 1H), 7.97–7.91 (m, 2H), 7.65–7.51 (m, 4H), 4.61–4.45 (m, 4H), 3.40–3.26 (m, 4H), 2.94 (s, 6H), 2.92 (s, 6H), 2.91 (s, 3H), 2.37 (br, 5H). ^13^C NMR (salt, CD_3_OD, 75 MHz): *δ* 152.1, 141.9, 136.1, 132.3, 131.7, 131.6, 131.3, 131.1, 130.6, 130.2, 129.8, 129.6, 122.2, 120.9, 118.3, 62.1, 60.9, 55.6, 53.9, 43.6, 43.1, 40.2, 21.1, 10.4. MS (ESI) *m*/*z* 420 [M + H]^+^. P_HPLC_ > 99%. HPLC (C_4_, 35 min): *t_R_* 12.8 min, P_HPLC_ 100%; HPLC (C_18_, 35 min): *t_R_* 17.9 min, P_HPLC_ 99%.

#### 4.1.68. {[4-(1-{3-[(Dimethylamino)methyl]phenyl}-4-methyl-1H-pyrazol-3-yl)phenyl] methyl} [3-(Dimethylamino)propyl]methylamine (**67**)

General procedure E: 63% yield. The compound was converted to its 3HCl salt (white solid). ^1^H NMR (free amine, CD_3_OD, 300 Mz): *δ* 8.05–8.05 (m, 1H), 7.75–7.67 (m, 4H), 7.44–7.39 (m, 3H), 7.23–7.21 (m, 1H), 3.54 (s, 2H), 3.51 (s, 2H), 2.41 (t, *J* = 7.4 Hz, 2H), 2.36–2.31 (m, 2H), 2.29–2.22 (m, 18H), 1.77–1.67 (m, 2H). ^13^C NMR (free amine, CD_3_OD, 75 MHz): *δ* 152.7, 141.4, 140.7, 138.9, 133.9, 130.6, 130.5, 129.2, 128.6, 128.2, 120.6, 118.7, 117.5, 64.7, 62.9, 58.6, 56.2, 45.4, 45.3, 42.4, 25.7, 10.4. MS (ESI) *m*/*z* 420 [M + H]^+^. P_HPLC_ > 98%. HPLC (C_4_, 30 min): *t_R_* 12.6 min, P_HPLC_ 98%; HPLC (C_18_, 30 min): *t_R_* 16.6 min, P_HPLC_ 100%.

#### 4.1.69. [3-(Dimethylamino)propyl](methyl){[4-(4-methyl-1-{3-[(4-methylpiperazin-1-yl) methyl]phenyl}-1H-pyrazol-3-yl)phenyl]methyl}amine (**68**)

General procedure E: 49% yield. The compound was converted to its 4HCl salt (white solid). ^1^H NMR (free amine, CD_3_OD, 300 Mz): *δ* 8.06–8.06 (m, 1H), 7.78–7.77 (m, 1H), 7.74–7.71 (m, 2H), 7.69–7.65 (m, 1H), 7.45–7.39 (m, 3H), 7.26–7.24 (m, 1H), 3.59 (s, 2H), 3.57 (s, 2H), 2.51 (br, 8H), 2.46–2.41 (m, 2H), 2.40–2.34 (m, 2H), 2.30–2.24 (m, 15H), 1.79–1.69 (m, 2H). ^13^C NMR (free amine, CD_3_OD, 75 MHz): *δ* 152.8, 141.4, 140.4, 138.9, 134.0, 130.7, 130.5, 129.3, 128.7, 128.3, 120.7, 118.7, 117.6, 63.5, 62.9, 58.6, 56.2, 55.7, 53.6, 46.0, 45.3, 42.4, 25.6, 10.3. MS (ESI) *m*/*z* 475 [M + H]^+^. P_HPLC_ > 99%. HPLC (C_4_, 30 min): *t_R_* 11.7 min, P_HPLC_ 100%; HPLC (C_18_, 30 min): *t_R_* 16.2 min, P_HPLC_ 99%.

#### 4.1.70. [3-(Dimethylamino)propyl][(4-{1-[3-({[3-(dimethylamino)propyl](methyl) amino}methyl)phenyl]-4-methyl-1H-pyrazol-3-yl}phenyl)methyl]methylamine (**69**)

General procedure E: 59% yield. The compound was converted to its 4HCl salt (white solid). ^1^H NMR (free amine, CD_3_OD, 300 Mz): *δ* 8.06–8.06 (m, 1H), 7.77–7.76 (m, 1H), 7.73 (d, *J* = 8.2 Hz, 2H), 7.68–7.64 (m, 1H), 7.44–7.38 (m, 3H), 7.25–7.22 (m, 1H), 3.57 (s, 2H), 3.55 (s, 2H), 2.42 (t, *J* = 7.4 Hz, 4H), 2.37–2.29 (m, 7H), 2.24–2.22 (m, 18H), 1.78–1.67 (m, 4H). ^13^C NMR (free amine, CD_3_OD, 75 MHz): *δ* 152.7, 141.4, 141.4, 139.0, 134.0, 130.6, 130.4, 129.2, 128.6, 128.1, 120.5, 118.5, 117.5, 62.9, 62.9, 58.6 (2C), 56.2 (2C), 45.4 (2C), 42.5, 42.4, 25.7, 25.7, 10.4. MS (ESI) *m*/*z* 491 [M + H]^+^. P_HPLC_ > 98%. HPLC (C_4_, 30 min): *t_R_* 11.9min, P_HPLC_ 98%; HPLC (C_18_, 30 min): *t_R_* 15.9 min, P_HPLC_ 98%.

#### 4.1.71. *N*′-[[3-[1-[3-(Dimethylaminomethyl)phenyl]-4-methyl-pyrazol-3-yl]phenyl] Methyl]-*N*,*N*,*N*′-trimethyl-propane-1,3-diamine (**70**)

General procedure E: 84% yield. The compound was converted to its 3HCl salt (white solid). ^1^H NMR (free amine, CD_3_OD, 300 Mz): *δ* 8.05 (s, 1H), 7.76–7.73 (m, 2H), 7.71–7.64 (m, 2H), 7.44–7.39 (m, 2H), 7.40 (ddd, *J* = 7.7, 1.3, 1.3 Hz, 1H), 7.23–7.21 (m, 1H), 3.57 (s, 2H), 3.51 (s, 2H), 2.46–2.35 (m, 4H), 2.29–2.24 (m, 18H), 1.78–1.68 (m, 2H). ^13^C NMR (free amine, CD_3_OD, 75 MHz): *δ* 152.8, 141.4, 140.7, 139.7, 135.0, 130.5, 129.9, 129.7, 129.5, 129.2, 128.2, 127.7, 120.6, 118.7, 117.6, 64.7, 63.1, 58.6, 56.2, 45.3 (2C), 42.4, 25.5, 10.4. MS (ESI) *m*/*z* 420 [M + H]^+^. P_HPLC_ > 96%. HPLC (C_4_, 35 min): *t_R_* 7.9 min, P_HPLC_ 99%; HPLC (C_18_, 35 min): *t_R_* 12.7 min, P_HPLC_ 96%.

#### 4.1.72. *N*,*N*,*N*′-Trimethyl-*N*′-[[3-[4-methyl-1-[3-[(4-methylpiperazin-1-yl)methyl] phenyl]pyrazol-3-yl]phenyl]methyl]propane-1,3-diamine (**71**)

General procedure E: 89% yield. The compound was converted to its 4HCl salt (white solid). ^1^H NMR (free amine, CD_3_OD, 300 Mz): *δ* 8.05 (s, 1H), 7.78–7.73 (m, 2H), 7.69–7.64 (m, 2H), 7.44–7.38 (m, 2H), 7.33–7.31 (m, 1H), 7.25–7.22 (m, 1H), 3.57 (s, 2H), 3.56 (s, 2H), 2.49 (br, 8H), 2.45–2.33 (m, 4H), 2.29–2.23 (m, 15H), 1.77–1.67 (m, 2H). ^13^C NMR (free amine, CD_3_OD, 75 MHz): *δ* 152.8, 141.4, 140.5, 139.7, 135.0, 130.4, 129.9, 129.7, 129.5, 129.2, 128.2, 127.7, 120.6, 118.7, 117.6, 63.5, 63.1, 58.6, 56.2, 55.7, 53.6, 46.0, 45.4, 42.5, 25.6, 10.4. MS (ESI) *m*/*z* 475 [M + H]^+^. P_HPLC_ > 99%. HPLC (C_4_, 35 min): *t_R_* 4.5 min, P_HPLC_ 100%; HPLC (C_18_, 35 min): *t_R_* 12.5 min, P_HPLC_ 99%.

#### 4.1.73. *N*′-[[3-[1-[3-[[3-(Dimethylamino)propyl-methyl-amino]methyl]phenyl]-4-methyl-pyrazol-3-yl]phenyl]methyl]-*N*,*N*,*N*′-trimethyl-propane-1,3-diamine (**72**)

General procedure E: 66% yield. The compound was converted to its 4HCl salt (white solid). ^1^H NMR (free amine, CD_3_OD, 300 Mz): *δ* 8.07 (s, 1H), 7.78–7.73 (m, 2H), 7.69–7.65 (m, 2H), 7.42 (dd, *J* = 7.7, 7.7 Hz, 2H), 7.35–7.32 (m, 1H), 7.26–7.24 (m, 1H), 3.59 (s, 2H), 3.58 (s, 2H), 2.47–2.41 (m, 4H), 2.39–2.34 (m, 4H), 2.3–2.24 (m, 21H), 1.79–1.69 (m, 4H). ^13^C NMR (free amine, CD_3_OD, 75 MHz): *δ* 152.9, 141.4, 139.7, 135.0, 130.4, 129.9, 129.8, 129.5, 129.2, 128.1, 127.7, 120.6, 118.6, 117.6, 63.1, 62.9, 58.6 (2C), 56.3, 56.2, 45.4 (2C), 42.5 (2C), 25.7 (2C), 10.4. MS (ESI) *m*/*z* 491 [M + H]^+^. P_HPLC_ > 98%. HPLC (C_4_, 35 min): *t_R_* 7.7 min, P_HPLC_ 99%; HPLC (C_18_, 35 min): *t_R_* 12.1 min, P_HPLC_ 98%.

### 4.2. Biological Evaluations

#### 4.2.1. Cytotoxicity Assays

Cytotoxicity was measured using 3-(4,5-dimethylthiazol-2-yl)-5-(3-carboxymethoxyphenyl)-2-(4-sulfophenyl)-2H-tetrazolium (MTS) tests. SY5Y-APP^695WT^ cells were plated in a 96-well plate at 3 × 10^4^ cells per well and allowed to attach for 24 h. Cells were then incubated for 72 h with 100 µL of DMEM with 10% SVF containing (or not) the defined concentration of drugs. Cytotoxicity was determined by using the colorimetric MTS assay (Cell Titer 96^®^ Aqueous One Solution Cell Proliferation Assay-MTS Promega, Madison, WI, USA) according to the manufacturer’s instructions. Absorbance was read at 490 nm.

#### 4.2.2. Cell Culture and Treatments

The human neuroblastoma cell line SY5Y-APP^695WT^ was maintained in Dulbecco’s modified Eagle medium (DMEM, high glucose, pyruvate—GIBCO, Life Technologies, Carlsbad, CA, USA) supplemented with 10% fetal bovine serum, 2 mM L-glutamine, 1 mM non-essential amino acids and penicillin/streptomycin (GIBCO, Life Technologies, Carlsbad, CA, USA) at 37 °C in a 5% CO_2_ humidified incubator [36]. For compound treatment, a 10 mM stock solution was diluted in freshly supplemented DMEM medium to obtain the precise final concentration of the drug. Cells were plated at a density of 5 × 10^5^ cells per well into 12-well plates and cultured with 1 mL supplemented DMEM cell medium for 24 h before compound exposure. The following day, the cell medium was replaced with fresh medium containing the compounds diluted at the indicated concentrations. Cells were treated for 24 h. At the end of the treatments, the cell medium was collected and kept at −80 °C until use, cells were rinsed once with PBS and extracted in 100 µL of Laemmli buffer (10 mM Tris, 20% glycerol, and 2% sodium dodecyl sulfate) using a cell-scraper. The cell lysate was further sonicated (30 pulses of 0.5 s, 60 Hz) for 5 min. Total protein concentration was determined using the Pierce BCA Protein Assay Kit (Thermo Scientific, Waltham, MA, USA) according to the manufacturer’s instructions. Samples were stored at −80 °C until analysis.

#### 4.2.3. Western Blot Analysis

Cell protein lysates were prepared for Western blot analysis by diluting the sample with 1 volume of NuPAGE^®^ lithium dodecyl sulfate (LDS) 2X sample buffer supplemented with 20% NuPAGE^®^ sample reducing agents (Invitrogen). Samples were heated for 10 min at 100 °C. Ten µg of total proteins per well were loaded onto precast 4–12% Criterion XT Bis-Tris polyacrylamide gels (Bio-Rad) and electrophoresis was achieved after applying a tension of 150 V for 90 min using a Criterion electrophoresis cell with the NuPAGE^®^ MOPS SDS running buffer (1X). Proteins were transferred to a nitrocellulose membrane of 0.45 µM pore size (G&E Healthcare, Madison, WI, USA) using the Criterion blotting system and applying a tension of 100 V for 45 min. To resolve proteins of low molecular weights such as carboxy-terminal fragments of APP, 12% Criterion XT Bis-Tris polyacrylamide gels (Bio-Rad, Hercules, CA, USA) were used, and electrophoresis was performed for 70 min at 150 V in a NuPAGE^®^ MES SDS running buffer (1X). Proteins were transferred to a nitrocellulose membrane of 0.2 µm pore size (G&E Healthcare, Madison, WI, USA) at 100 V for 40 min. Molecular weight calibration was achieved using molecular weight markers (Novex and Magic Marks, Life Technologies, Carlsbad, CA, USA). Protein transfer and quality were determined by a reversible Ponceau Red coloration (0.2% xylidine Ponceau Red and 3% trichloroacetic acid). Membranes were then blocked in 25 mM Tris-HCl pH 8.0, 150 mM NaCl, 0.1% Tween-20 (*v*/*v*) (TNT), and 5% (*w*/*v*) of skimmed milk or 5% (*w*/*v*) of bovine serum albumin depending on the antibody during 1 h. The membrane was rinsed three times in TNT for 10 min before incubation with the primary antibody overnight at 4 °C. The membrane was incubated with the secondary antibody for 45 min at rt. The immunoreactive complexes were revealed using the ECL^TM^ Western Blotting Detection Reagents (G&E Healthcare, Madison, WI, USA) and image acquisitions were performed with the Amersham Imager 600 (G&E Healthcare, Madison, WI, USA). Quantifications of the protein expression levels were performed with Image Quant TL (G&E Healthcare, Madison, WI, USA).

#### 4.2.4. Antibodies

Primary antibodies used in this study for Western blot analysis included a well-characterized homemade rabbit antiserum against the last 17 amino acids of APP, named APP-Cter-C17 (1/5000) [11,12,37], LC3B obtained from Cell Signaling (1/1000), p62 (Abcam, 1/2000, Cambridge, GB), and α-tubulin (Sigma, 1/10,000). The anti-histone H3 (1/10,000) used for normalization was obtained from Sigma (Saint-Louis, MO, USA). Secondary antibodies (peroxidase-labeled goat anti-rabbit IgG, 1/5000 or peroxidase-labeled horse anti-mouse IgG, 1/50,000) were obtained from Vector Laboratories (Eurobio Scientific, Les Ulis, France).

#### 4.2.5. Quantification of Secreted Aβ and sAPP

Conditioned media of SY5Y-APP^695WT^ collected at the end of treatments were centrifuged at 1000× *g* for 5 min to eliminate cell debris.

Aβ_1–40_/Aβ_1–42_ peptides, Aβ_x–38_/Aβ_x–40_/Aβ_x–42_, and sAPPα/sAPPβ concentrations in pg/mL were determined, respectively, using amyloid-beta 40 and 42 Human ELISA Kits (Invitrogen), the V-PLEX Plus Aβ Peptide Panel 1 (4G8) Kit (Meso Scale Diagnostics, MSD R©, Rockville, MD, USA), and the sAPPα/sAPPβ Multiplex Kit (Meso Scale Diagnostics, MSD R©, Rockville, MD, USA) according to the manufacturer’s instructions.

#### 4.2.6. Statistical Analysis

Statistical analyses were performed using ANOVA followed by Fisher’s LSD test. All tests were performed by using GraphPad Prism 9.4.1 (GraphPad Software, San Diego, CA, USA) and statistical significance was set at * *p* < 0.05, ** *p* < 0.001, *** *p* < 0.001, and **** *p* < 0.0001. All data are reported as mean ± SD.

## 5. Patents

The results have been patented (EP22306550.9, 12 October 2022).

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
