# Peer review of "Discovery of Compounds That Selectively Repress the Amyloidogenic Processing of the Amyloid Precursor Protein: Design, Synthesis and Pharmacological Evaluation of Diphenylpyrazoles"

_ijms, 2022, doi:10.3390/ijms232113111_

Round 1
Reviewer 1 Report
P. Melnyk and co-workers describe a pharmacophore-based design, synthesis, and pharmacological evaluation of some 1,3-diphenylpyrazoles with potential anti-amyloidogenic properties, and hence, with clear potential to be further in-depth evaluated behind understanding their plausible mechanisms and modes of action. Main findings support that assayed compounds may be considered as anti-Alzheimer drug candidates. Thus, I found enough positive aspects making this work publishable in the IJMS. For instance, enough novelty, originality, robustness, and relevance come from this work, the study was well conducted, references cited are pertinent, results are described in detail, discussion of results is well supported, grammar, style and format are Okay, and experimental part makes the work reproducible, among other interesting features. However, nowadays it is mandatory to submit a supplementary material along with a serious manuscript. Thus, I would recommend publication of this manuscript only if Authors submit a satisfactory complementary file including spectra of all new compounds, I mean, 1H and 13C NMR (1D and 2D), MS, and HPLC chromatograms. It should not be any problem.
Author Response
Melnyk and co-workers describe a pharmacophore-based design, synthesis, and pharmacological evaluation of some 1,3-diphenylpyrazoles with potential anti-amyloidogenic properties, and hence, with clear potential to be further in-depth evaluated behind understanding their plausible mechanisms and modes of action. Main findings support that assayed compounds may be considered as anti-Alzheimer drug candidates. Thus, I found enough positive aspects making this work publishable in the IJMS. For instance, enough novelty, originality, robustness, and relevance come from this work, the study was well conducted, references cited are pertinent, results are described in detail, discussion of results is well supported, grammar, style and format are Okay, and experimental part makes the work reproducible, among other interesting features.
We thank the reviewer for its positive comments
However, nowadays it is mandatory to submit a supplementary material along with a serious manuscript. Thus, I would recommend publication of this manuscript only if Authors submit a satisfactory complementary file including spectra of all new compounds, I mean, 1H and 13C NMR (1D and 2D), MS, and HPLC chromatograms. It should not be any problem.
Entire spectra and chromatograms are added in a 181-page document as supplementary raw materials
Reviewer 2 Report
The article deals with the discovery of compounds that selectively repress the amyloidogenic processing of the amyloid precursor protein (APP). As objects of study, a series of 1,4-diphenyl-1H-pyrazoles with dialkyl(amino) side chains were selected. Synthetic strategies leading to the target compounds, along with their effects on the metabolism of APP and cytotoxicity evaluation, have been presented. As a result, three compounds with acceptable activitiy and cytotoxicity parameters were identified as the lead ones.
Overall, the article makes a favorable impression, but some comments should nevertheless be made thereon.
1. No explanation was given as to why 1H-pyrazole derivatives were those that were selected for study.
2. The selection of a side chain occupying one of the phenyl ring positions appears to be random. Anyway, no comment regarding it was given.
3. Note that only high resolution mass spectra may exclude the need to provide elemental analysis data. Low resolution ones may not, unfortunately.
Author Response
The article deals with the discovery of compounds that selectively repress the amyloidogenic processing of the amyloid precursor protein (APP). As objects of study, a series of 1,4-diphenyl-1H-pyrazoles with dialkyl(amino) side chains were selected. Synthetic strategies leading to the target compounds, along with their effects on the metabolism of APP and cytotoxicity evaluation, have been presented. As a result, three compounds with acceptable activitiy and cytotoxicity parameters were identified as the lead ones.
Overall, the article makes a favorable impression, but some comments should nevertheless be made thereon.
- No explanation was given as to why 1H-pyrazole derivatives were those that were selected for study.
Starting from chloroquine and amodiaquine, which demonstrated a strong inhibitory effect on both Aβ1-40 and Aβ1-42 secretions and a strong increase of αCTFs and AICD levels, families A and B were developed and gave similar interesting biological effects. Following these studies, a computer-assisted ligand-based approach, with derivatives of the two families A and B, allowed to determine a common pharmacophoric model (Eur. J. Med. Chem. 2018, 159, 104-125). Indeed, several structures were then proposed to spacially organize pharmacophoric elements in agreement with this model. We designed structures based on a scaffold enabling the orinetation of amino side chains in three directions. A first series with a biaryl scaffold was developed (family C already published : Eur. J. Med. Chem. 2018, 159, 104-125). We conceptualized other scaffolds derived from several heterocycles such as benzimidazole. The latter were however showing lower biological activity and thus out of the scope of the present series of compounds. We focused here our efforts on diphenyl pyrazole scaffold (family D reported in the present manuscript).
The following sentence (in blue in the text) was added to help the understanding of the paper (line 121) :
Several structures were then conceptualized to spatially organize pharmacophoric elements in agreement with this model. We then choose chemical structures based on a scaffold enabling the orientation of amino side chains in three different orientations.
- The selection of a side chain occupying one of the phenyl ring positions appears to be random. Anyway, no comment regarding it was given.
The nature of the side chains on the pyrazole scaffold was choosen according to previous research and biological results of our group (ACS Chem. Neurosci. 2015, 6, 559–569, Bioorganic Med. Chem. 2018, 26, 2151–2164, Eur. J. Med. Chem. 2018, 159, 104-125, Neurobiol. Dis. 2019, S0969-9961, 30327-9). These different modulations allowed us to explore, and validate the importance of the 2 amino side chains (with methyl group at the 4 position of the pyrazole) or the 3 amino side chains. The nature of each side chains allowed us to determine the best biological profil for compounds with pyrazole scaffold.
The following sentence (in blue in the text) was added to help the understanding of the paper (line 121) :
The nature of the amino side chains on the pyrazole scaffold was chosen based on previous biological readouts. Different modulations then allowed us to select and validate the importance of adding 2 or 3 different amino side chains: monoamine, linear or cyclic diamine.
3. Note that only high resolution mass spectra may exclude the need to provide elemental analysis data. Low resolution ones may not, unfortunately.
We agree with the reviewer. In a majority of journal dedicated to medicinal chemistry, high resolution mass spectra or elemental analyses are not mandatory and we don’t have this facility in Lille. To circomvent this drawback, for all final compounds, the purity was verified by two types of high-pressure liquid chromatography (HPLC) columns: C18 Interchrom UPTISPHERE and C4 Interchrom, with a long 35 min running method . All final compounds displayed purities of more than 95%. 1H and 13C NMR spectra have been recorded. All the spectra and chromatograms have been now added in a 181-page document as supplementary raw materials.
Round 2
Reviewer 1 Report
The Authors attended my suggestion about submitting an Electronic Supplementary Material along with the revised version of the manuscript. Thus, I recommend publication of this work in its present form.